# JoMA: Demystifying Multilayer Transformers via JOint Dynamics of MLP and Attention

**Yuandong Tian**
AI@Meta (FAIR)
yuandong@meta.com

**Yiping Wang**
University of Washington
ypwang61@cs.washington.edu

**Zhenyu Zhang**
University of Texas at Austin
zhenyu.zhang@utexas.edu

**Beidi Chen**
Carnegie Mellon University, AI@Meta (FAIR)
beidic@meta.com, beidic@andrew.cmu.edu

**Simon Du**
University of Washington
ssdu@cs.washington.edu

## Abstract

We propose **Jo**int **MLP**/**A**ttention (JoMA) dynamics, a novel mathematical framework to understand the training procedure of multilayer Transformer architectures. This is achieved by *integrating out* the self-attention layer in Transformers, producing a modified dynamics of MLP layers only. JoMA removes unrealistic assumptions from previous analysis (e.g., lack of residual connection) and predicts that the attention first becomes sparse (to learn salient tokens), then dense (to learn less salient tokens) in the presence of nonlinear activations, while in the linear case, it is consistent with existing works that show attention becomes sparse over time. We leverage JoMA to qualitatively explains how tokens are combined to form hierarchies in multilayer Transformers, when the input tokens are generated by a latent hierarchical generative model. Experiments on models trained from real-world dataset (Wikitext2/Wikitext103) and various pre-trained models (OPT, Pythia) verify our theoretical findings. The code is at[1].

## 1 Introduction

Since its debut, Transformers (Vaswani et al., 2017) have been extensively used in many applications and demonstrates impressive performance (Dosovitskiy et al., 2020; OpenAI, 2023) compared to domain-specific models (e.g., CNN in computer vision, GNN in graph modeling, RNN/LSTM in language modeling, etc). In all these scenarios, the *basic Transformer block*, which consists of **one self-attention plus two-layer nonlinear MLP**, plays a critical role. A natural question arises:

*How the basic Transformer block leads to effective learning?*

Due to the complexity and nonlinearity of Transformer architectures, it remains a highly nontrivial open problem to find a unified mathematical framework that characterizes the learning mechanism of *multi-layer* transformers. Existing works mostly focus on 1-layer Transformer (Li et al., 2023a; Tarzanagh et al., 2023b) with fixed MLP (Tarzanagh et al., 2023a) layer, linear activation functions (Tian et al., 2023), and local gradient steps at initialization (Bietti et al., 2023; Oymak et al., 2023), etc.

In this paper, we propose a novel joint dynamics of self-attention plus MLP, based on **Jo**int **MLP**/**A**ttention Integral (JoMA), a first integral that combines the lower layer of the MLP and self-attention layers. Leveraging this joint dynamics, the self-attention is shown to have more fine-grained and delicate behavior: it first becomes *sparse* as in the linear case (Tian et al., 2023), only attends to tokens that frequently co-occur with the query, and then becomes *denser* and gradually includes tokens with less frequent co-occurrence, in the case of nonlinear activation. This shows a *changing* inductive bias in the Transformer training: first the model focuses on most salient features, then extends to less salient ones.

Another natural question arises: why such a learning pattern is preferred? While for 1-layer this does not give any benefits, in multilayer Transformer setting, we show qualitatively that such a dynamics plays an important role. To demonstrate that this is the case, we assume a hierarchical tree generative model for the input tokens. In this model, starting from the

---

[1] https://github.com/facebookresearch/luckmatters/tree/yuandong3

upper level latent variables (in which the top-most is the class label of the input sequence), abbreviated as $\text{LV}_s$, generates the latents $\text{LV}_{s-1}$ in the lower layer, until reaching the token level ($s = 0$). With this model, we show that the tokens generated by the lowest latents $\text{LV}_1$ co-occur a lot and thus can be picked up first by the attention dynamics as "salient features". This leads to learning of such token combinations in hidden MLP nodes, which triggers self-attention grouping at $s = 1$, etc. In this way, the non-salient co-occurrences are naturally explained by the top level hierarchy, rather than incorrectly learned by the lower layer as spurious correlation, which is fortunately delayed by the attention mechanism. Our theoretical finding is consistent with both the pre-trained models such as OPT/Pythia and models trained from scratch using real-world dataset (Wikitext2 and Wikitext103).

We show that JoMA overcomes several main limitations from Scan&Snap (Tian et al., 2023). JoMA incorporates residual connections and MLP nonlinearity as key ingredients, analyzes joint training of MLP and self-attention layer, and qualitatively explains dynamics of multi-layer Transformers. For linear activation, JoMA coincides with Scan&Snap, i.e., the attention becomes sparse during training.

## 1.1 RELATED WORK

**Training Dynamics of Neural Networks.** Earlier research has delved into training dynamics within multi-layer linear neural networks (Arora et al., 2018; Bartlett et al., 2018), the teacher-student setting (Brutzkus & Globerson, 2017; Tian, 2017; Soltanolkotabi, 2017; Du et al., 2017; 2018a; Xu & Du, 2023), and infinite-width limits (Jacot et al., 2018; Du et al., 2018b; Allen-Zhu et al., 2019; Oymak & Soltanolkotabi, 2020; Li & Liang, 2018; Nguyen & Pham, 2020; Fang et al., 2021). This includes extensions to attention-based-models (Hron et al., 2020; Yang et al., 2022). In self-supervised learning, analysis exists for dynamics in deep linear networks (Tian, 2022) and the impact of nonlinearity (Tian, 2023).

**Dynamics for Attention-based models**. Zhang et al. (2020) delves into adaptive optimization techniques. Jelassi et al. (2022) demonstrates that the vision transformer (Dosovitskiy et al., 2020) trained via gradient descent can discern spatial structures. Li et al. (2023c) illustrates that a single-layer Transformer can learn a constrained topic model, where each word is tied to a single topic, using $\ell_2$ loss, BERT-like framework (Devlin et al., 2018), and certain assumptions on attention patterns. Snell et al. (2021) investigate the training dynamics of single-head attention in mimicking Seq2Seq learning. Tian et al. (2023) characterizes the SGD training dynamics of a 1-layer Transformer and shows that with cross-entropy loss, the model will pay more attention to the key tokens that frequently co-occur with the query token. Oymak et al. (2023) constructs the attention-based contextual mixture model and demonstrates how the prompt can attend to the sparse context-relevant tokens via gradient descent. Tarzanagh et al. (2023b) also finds that running gradient descent will converge in direction to the max-margin solution that separates the locally optimal tokens from others, and Tarzanagh et al. (2023a) further disclose the connection between the optimization geometry of self-attention and hard-margin SVM problem. For the in-context learning scenario, several recent works analyze linear transformers trained on random instances for linear regression tasks from the perspective of loss landscape (Boix-Adsera et al., 2023; Zhang et al., 2023). While these studies also study the optimization dynamics of attention-based models, they do not reveal the phenomena that we discuss.

**Expressiveness of Attention-based Models**. The universal approximation abilities of attention-based models have been studied extensively (Yun et al., 2019; Bhattamishra et al., 2020a;b; Dehghani et al., 2018; Pérez et al., 2021). More recent studies offer detailed insights into their expressiveness for specific functions across various scenarios, sometimes incorporating statistical evaluations (Edelman et al., 2022; Elhage et al., 2021; Likhosherstov et al., 2021; Akyürek et al., 2022; Zhao et al., 2023; Yao et al., 2021; Anil et al., 2022; Barak et al., 2022). A fruitful line of work studied the in-context learning capabilities of the Transformer (Dong et al., 2022), linking gradient descent in classification/regression learning to the feedforward actions in Transformer layers (Garg et al., 2022; Von Oswald et al., 2022; Bai et al., 2023; Olsson et al., 2022; Akyürek et al., 2022; Li et al., 2023b). However, unlike our study, these work do not characterize the training dynamics.

## 2 PROBLEM SETTING

Let the total vocabulary size be $M$, in which $M_C$ is the number of contextual tokens and $M_Q$ is the number of query tokens. Consider one layer in multilayer transformer (Fig. 1(b)):

$$h_k = \phi(\boldsymbol{w}_k^\top \boldsymbol{f}), \quad \boldsymbol{f} = U_C \boldsymbol{b} + \boldsymbol{u}_q, \quad \boldsymbol{b} = \sigma(\boldsymbol{z}_q) \circ \boldsymbol{x}/A \tag{1}$$

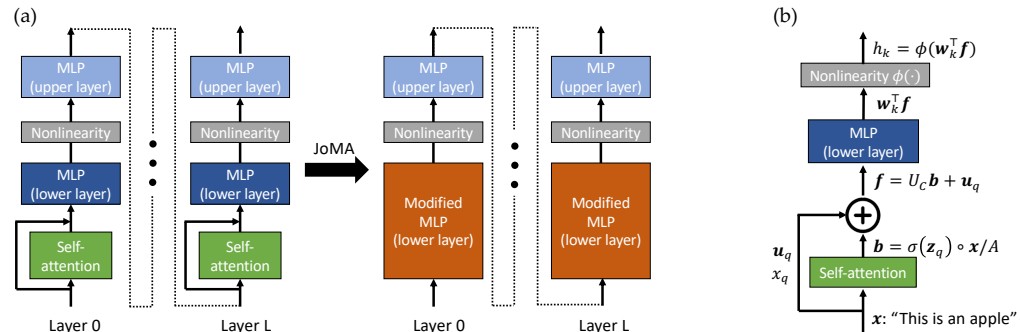

Figure 1: **(a)** Overview of JoMA framework. Using the invariant of training dynamics, the self-attention layer and the lower layer of MLP can be merged together to yield a MLP layer with modified dynamics (Theorem 1), which explains the behaviors of attention in linear (Sec. 3.1) and nonlinear (Sec. 4) MLP activation $\phi$, as well as hierarchical concept learning in multilayer cases (Sec. 5). **(b)** Problem setting. JoMA frameworks support different kind of attentions, including linear attention $b_l := x_l z_{ql}$, exp attention $b_l := x_l e^{z_{ql}}/A$ and softmax $b_l := x_l e^{z_{ql}} / \sum_l x_l e^{z_{ql}}$.

**Input/outputs**. $\boldsymbol{x} = [x_l] \in \mathbb{R}^{M_C}$ is the input frequency vector for contextual token $1 \leq l \leq M_C$, $1 \leq q \leq M_Q$ is the query token index, $K$ is the number of nodes in the hidden MLP layer, whose outputs are $h_k$. All the quantities above vary across different sample index $i$ (i.e., $x_l = x_l[i]$, $q = q[i]$). In addition, $\phi$ is the nonlinearity (e.g., ReLU).

**Model weights**. $\boldsymbol{z}_q = [z_{ql}] \in \mathbb{R}^{M_C}$ is the (unnormalized) attention logits given query $q$, and $\boldsymbol{w}_k \in \mathbb{R}^d$ are the weights for the lower MLP layer. These will be analyzed in the paper.

**The Attention Mechanism**. In this paper, we mainly study three kinds of attention:

- *Linear Attention (Von Oswald et al., 2022)*: $\sigma(x) = x$ and $A := 1$;

- *Exp Attention*: $\sigma(x) = \exp(x)$ and $A := \text{const}$;

- *Softmax Attention (Vaswani et al., 2017)*: $\sigma(x) = \exp(x)$ and $A := \mathbf{1}^\top (\sigma(\boldsymbol{z}_q) \circ \boldsymbol{x})$.

Here $\circ$ is the Hadamard (element-wise) product. $\boldsymbol{b} \in \mathbb{R}^{M_C}$ are the attention scores for contextual tokens, given by a point-wise *attention function* $\sigma$. $A$ is the normalization constant.

**Embedding vectors**. $\boldsymbol{u}_l$ is the embedding vector for token $l$. We assume that the embedding dimension $d$ is sufficiently large and thus $\boldsymbol{u}_l^\top \boldsymbol{u}_{l'} = \mathbb{I}(l = l')$, i.e., $\{\boldsymbol{u}_l\}$ are orthonormal bases. Let $U_C = [\boldsymbol{u}_1, \boldsymbol{u}_2, \ldots, \boldsymbol{u}_{M_C}] \in \mathbb{R}^{d \times M_C}$ be the matrix that encodes all embedding vectors of contextual tokens. Then $U_C^\top U_C = I$. Appendix B.1 verifies the orthogonality assumption in multiple pre-trained models (Pythia, LLaMA, etc).

**Residual connections** are introduced as an additional term $\boldsymbol{u}_q$ in Eqn. 1, which captures the critical component in Transformer architecture. Note that we do not model value matrix $W_V$ since it can be merged into the embedding vectors (e.g., by $\boldsymbol{u}_l' = W_V \boldsymbol{u}_l$), while $W_K$ and $W_Q$ are already implicitly modeled by the self-attention logits $z_{ql} = \boldsymbol{u}_q^\top W_Q^\top W_K \boldsymbol{u}_l$.

**Gradient backpropagation in multilayers**. In multilayer setting, the gradient gets backpropagated from top layer. Specifically, let $g_{h_k}[i]$ be the backpropagated gradient sent to node $k$ at sample $i$. For 1-layer Transformer with softmax loss directly applied to the hidden nodes of MLP, we have $g_{h_k}[i] \sim \mathbb{I}(y_0[i] = k)$, where $y_0[i]$ is the label to be predicted for sample $i$. For brevity, we often omit sample index $i$ if there is no ambiguity.

**Assumption 1** (Stationary backpropagated gradient $g_{h_k}$)**.** *Expectation terms involving $g_{h_k}$ (e.g., $\mathbb{E}[g_{h_k}\boldsymbol{x}]$) remains constant during training.*

Note that this is true for *layer-wise* training: optimizing the weights for a specific Transformer layer, while fixing the weights of others and thus the statistics of backpropagated are stationary. For joint training, this condition also holds approximately since the weights change gradually during the training process. Under Assumption 1, Appendix A.1 gives an equivalent formulation in terms of per-hidden node loss.

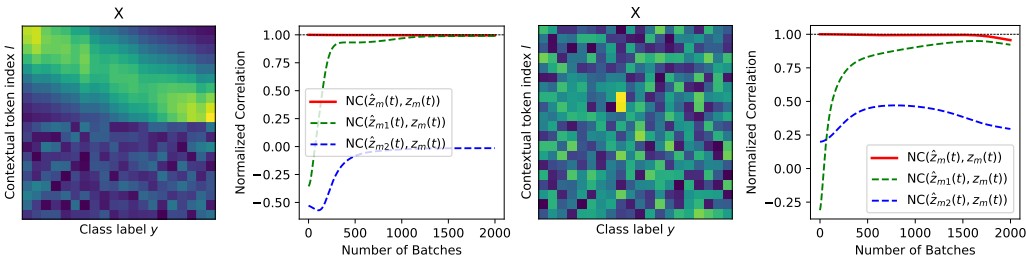

Figure 2: Test of training dynamics with linear MLP activation ($\phi(x) = x$) under softmax attention. **Left Two:** The distribution of $\boldsymbol{x}$ smoothly transits over different class labels. **Right Two:** The distribution of $\boldsymbol{x}$ over different classes are randomly generated. In both cases, the estimated $\hat{\boldsymbol{z}}_m(t)$ by the first integral (Theorem 1), despite assumptions on $\bar{\boldsymbol{b}}_m$, shows high correlation with the ground truth self-attention logits $\boldsymbol{z}_m(t)$, while its two components $\hat{\boldsymbol{z}}_{m1}(t) := \frac{1}{2}\sum_k \boldsymbol{v}_k^2(t)$ and $\hat{\boldsymbol{z}}_{m2}(t) := -\frac{1}{2}\sum_k \|\boldsymbol{v}_k(t)\|_2^2 \bar{\boldsymbol{b}}_m$ do not.

**Training Dynamics**. Define the conditional expectation $\mathbb{E}_{q=m}[\cdot] := \mathbb{E}[\cdot|q=m]$. Now let us consider the dynamics of $\boldsymbol{w}_k$ and $\boldsymbol{z}_m$, if we train the model with a batch of inputs that always end up with query $q[i] = m$, then:

$$\dot{\boldsymbol{w}}_k = \mathbb{E}_{q=m}\left[g_{h_k} h_k' \boldsymbol{f}\right], \qquad \dot{\boldsymbol{z}}_m = \mathbb{E}_{q=m}\left[(\partial \boldsymbol{b}/\partial \boldsymbol{z}_m)^\top U_C^\top \boldsymbol{g_f}\right] \tag{2}$$

Here $h_k' := \phi'(\boldsymbol{w}_k^\top \boldsymbol{f})$ is the derivative of current activation and $\boldsymbol{g_f} := \sum_k g_{h_k} h_k' \boldsymbol{w}_k$.

## 3 JoMA: Existence of JOint dynamics of Attention and MLP

While the learning dynamics of $\boldsymbol{w}_k$ and $\boldsymbol{z}_m$ can be complicated, surprisingly, training dynamics suggests that the attention logits $\boldsymbol{z}_m(t)$ have *close-form* relationship with respect to the MLP weights $\boldsymbol{w}_k(t)$, which lays the foundation of our JoMA framework:

**Theorem 1** (JoMA). *Let $\boldsymbol{v}_k := U_C^\top \boldsymbol{w}_k$, then the dynamics of Eqn. 2 satisfies the invariants:*

- *Linear attention. The dynamics satisfies $\boldsymbol{z}_m^2(t) = \sum_k \boldsymbol{v}_k^2(t) + \boldsymbol{c}$.*

- *Exp attention. The dynamics satisfies $\boldsymbol{z}_m(t) = \frac{1}{2}\sum_k \boldsymbol{v}_k^2(t) + \boldsymbol{c}$.*

- *Softmax attention. If $\bar{\boldsymbol{b}}_m := \mathbb{E}_{q=m}[\boldsymbol{b}]$ is a constant over time and $\mathbb{E}_{q=m}\left[\sum_k g_{h_k} h_k' \boldsymbol{b}\boldsymbol{b}^\top\right] = \bar{\boldsymbol{b}}_m \mathbb{E}_{q=m}\left[\sum_k g_{h_k} h_k' \boldsymbol{b}\right]$, then the dynamics satisfies $\boldsymbol{z}_m(t) = \frac{1}{2}\sum_k \boldsymbol{v}_k^2(t) - \|\boldsymbol{v}_k(t)\|_2^2 \bar{\boldsymbol{b}}_m + \boldsymbol{c}$.*

*Under zero initialization ($\boldsymbol{w}_k(0) = 0$, $\boldsymbol{z}_m(0) = 0$), then the time-independent constant $\boldsymbol{c} = 0$.*

Therefore, we don't need to explicitly update self-attention, since it is already implicitly incorporated in the lower layer of MLP weight! For softmax attention, we verify that even with the assumption, the invariance proposed by Theorem 1 still predicts $\boldsymbol{z}_m(t)$ fairly well.

### 3.1 Linear activations: winner-take-all

Now we can solve the dynamics of $\boldsymbol{w}_k(t)$ (Eqn. 2), by plugging in the close-form solution of self-attention. For simplicity, we consider exp attention with $K = 1$ (i.e., single hidden MLP node). Let $\Delta_m := \mathbb{E}_{q=m}[g_{h_k} h_k' \boldsymbol{x}]$, then $\boldsymbol{v}_k$'s dynamics is ($\boldsymbol{v}_k$ written as $\boldsymbol{v}$):

$$\dot{\boldsymbol{v}} = \Delta_m \circ \exp(\boldsymbol{z}_m) = \Delta_m \circ \exp(\boldsymbol{v}^2/2 + \boldsymbol{c}) \tag{3}$$

In the case of linear activations $\phi(x) = x$, $h_k' \equiv 1$. According to Assumption 1, $\Delta_m$ does not depend on $\boldsymbol{v}$ and we arrive at the following theorem:

**Theorem 2** (Linear Dynamics with Self-attention). *With linear MLP activation and zero initialization, for exp attention any two tokens $l \neq l'$ satisfy the following invariants:*

$$\frac{\mathrm{erf}\left(v_l(t)/2\right)}{\Delta_{lm}} = \frac{\mathrm{erf}\left(v_{l'}(t)/2\right)}{\Delta_{l'm}} \tag{4}$$

*where $\Delta_{lm} = \mathbb{E}_{q=m}[g_{h_k} x_l]$ and $\mathrm{erf}(x) = \frac{2}{\sqrt{\pi}}\int_0^x e^{-t^2} \mathrm{d}t$ is Gauss error function.*

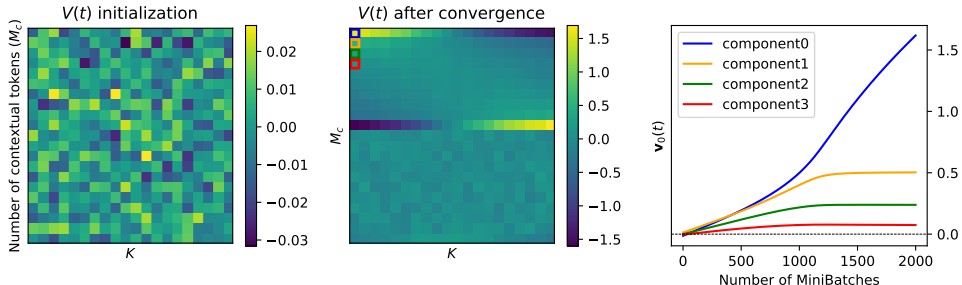

Figure 3: Growth of different components in $\boldsymbol{v}_0(t)$ (First few components of the first column of $V(t)$) in linear MLP activation and softmax attention. As predicted by Sec. 3.1, after convergence, only some components of $\boldsymbol{v}_0$ grows while the remaining components is saturated after initial growing, consistent with Theorem 2 even if it is derived from JoMA's approximation in Theorem 1. Each node $k$ (and thus $\boldsymbol{w}_k$) receives back-propagated gradient from $k$-th class via cross-entropy loss.

**Remarks.** The dynamics suggests that the weights become one-hot over training. Specifically, let $l^* = \arg\max_l |\Delta_{lm}|$, then $v_{l^*}(t) \to \mathrm{sign}(\Delta_{l^*m}) \times \infty$ and other $v_l(t)$ converges to finite numbers, because of the constraint imposed by Eqn. 4 (see Fig. 3). For softmax attention, there is an additional sample-dependent normalization constant $A[i]$, if $A[i]$ remains constant across samples and all elements of $\bar{\boldsymbol{b}}_m$ are the same, then Theorem 2 also applies.

**Beyond distinct/common tokens.** $\Delta_{lm} := \mathbb{E}_{l,q=m}[g_{h_k}]\mathbb{P}(l|m)^2$ is a product of *token discriminancy* (i.e., $\mathbb{E}_{l,q=m}[g_{h_k}] > 0$ means token $l$ positively correlated to backpropagated gradient $g_{h_k}$, or label in the 1-layer case) and *token frequency* (i.e., $\mathbb{P}(l|m)$, how frequent $l$ appears given $m$). This covers a broader spectrum of tokens than Tian et al. (2023), which only discusses distinct (i.e., large $|\Delta_{lm}|$) and common tokens (i.e., when $\Delta_{lm} \approx 0$).

## 4    Training Dynamics under Nonlinear Activations

In nonlinear case, the dynamics turns out to be very different. In this case, $\Delta_m$ is no longer a constant, but will change. As a result, the dynamics also changes substantially.

**Theorem 3** (Dynamics of nonlinear activation with uniform attention). *If $\boldsymbol{x}$ is sampled from a mixture of $C$ isotropic distributions centered at $[\bar{\boldsymbol{x}}_1, \ldots, \bar{\boldsymbol{x}}_C]$, where each $\bar{\boldsymbol{x}}_c \in \mathbb{R}^d$ and gradient $g_{h_k}$ are constant within each mixture, then:*

$$\dot{\boldsymbol{v}} = \Delta_m = \frac{1}{\|\boldsymbol{v}\|_2}\sum_c a_c\theta_1(r_c)\bar{\boldsymbol{x}}_c + \frac{1}{\|\boldsymbol{v}\|_2^3}\sum_c a_c\theta_2(r_c)\boldsymbol{v} \tag{5}$$

*here $a_c := \mathbb{E}_{q=m,c}[g_{h_k}]\mathbb{P}[c]$, $r_c := \boldsymbol{v}^\top\bar{\boldsymbol{x}}_c + \xi$ is the affinity to $\bar{\boldsymbol{x}}_c$ and the "bias" term $\xi(t) := \int_0^t \mathbb{E}_{q=m}[g_{h_k}h'_k]\,\mathrm{d}t$, $\theta_1$ and $\theta_2$ depend on derivative of nonlinearity $\psi := \phi'$ and data distribution but not $\boldsymbol{v}$. If $\psi$ is monotonous with $\psi(-\infty) = 0$ and $\psi(+\infty) = 1$, so does $\theta_1$.*

Appendix A.3.2 presents critical point analysis. Here we focus on a simplified one when $\boldsymbol{v}$ is constrained to be a unit vector, which leads to the following modified dynamics ($P_{\boldsymbol{v}}^\perp \boldsymbol{v} = 0$):

$$\dot{\boldsymbol{v}} = P_{\boldsymbol{v}}^\perp \Delta_m = \sum_c a_c\theta_1(r_c)P_{\boldsymbol{v}}^\perp\bar{\boldsymbol{x}}_c = \sum_c a_c\theta_1(r_c)\|\bar{\boldsymbol{x}}_c\|[\boldsymbol{\mu}_c - (\boldsymbol{v}^\top\boldsymbol{\mu}_c)\boldsymbol{v}] \tag{6}$$

where $\boldsymbol{\mu}_c := \bar{\boldsymbol{x}}_c/\|\bar{\boldsymbol{x}}_c\|$. We consider when $\boldsymbol{v}$ is aligned with one cluster $\bar{\boldsymbol{x}}_c$ but far away from others, then $r_c \gg r_{c'}$ for $c' \neq c$ and $\theta_1(r_c) \gg \theta_1(r_{c'})$ since $\theta_1$ is monotonously increasing. Hence $\boldsymbol{\mu}_c$ dominates and let $\boldsymbol{\mu} := \boldsymbol{\mu}_c$ for brevity. Similar to Eqn. 3, we use close-form simplification of JoMA to incorporate self-attention, which leads to (we use exp attention):

$$\dot{\boldsymbol{v}} \propto (\boldsymbol{\mu} - \boldsymbol{v}) \circ \exp(\boldsymbol{v}^2/2) \tag{7}$$

Here we omit the scalar terms and study when $\boldsymbol{v}$ is close to $\boldsymbol{\mu}$, in which $\boldsymbol{v}^\top\boldsymbol{\mu} = 1 + O(\|\boldsymbol{\mu} - \boldsymbol{v}\|_2^2) \approx 1$. It is clear that the critical point $\boldsymbol{v}_* = \boldsymbol{\mu}$ does not change after adding the term $\exp(\boldsymbol{v}^2/2)$. However, the convergence speed changes drastically. As shown in the following lemma, the convergence speed towards *salient* component of $\boldsymbol{\mu}$ (i.e., component with large magnitude) is much faster than non-salient ones:

---

[2]Since $x_l[i]$ is the empirical frequency of token $l$ in sample $i$, we have $\Delta_{lm} = \mathbb{E}_{q=m}[g_{h_k}x_l] = \sum_i g_{h_k}[i]\mathbb{P}(l|q=m,i)\mathbb{P}(i|q=m) = \sum_i g_{h_k}[i]\mathbb{P}(i|q=m,l)\mathbb{P}(l|q=m) = \mathbb{E}_{l,q=m}[g_{h_k}]\mathbb{P}(l|m)$.

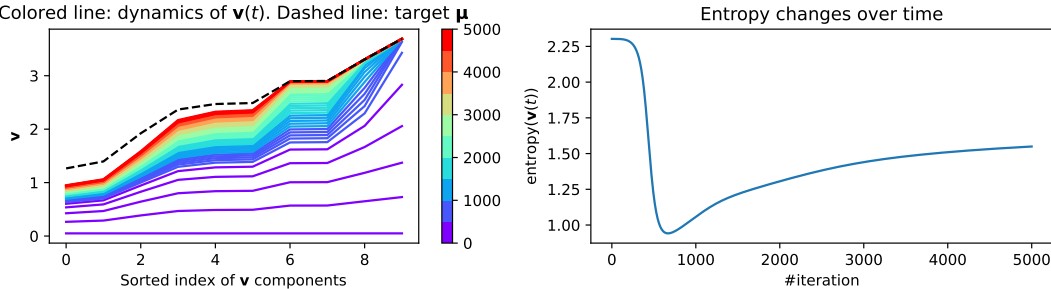

Figure 4: Dynamics of nonlinear MLP with self-attention components included (Eqn. 7). **Left:** Training dynamics (color indicating training steps). The salient components (i.e., components with large magnitude in $\boldsymbol{\mu}$) of $\boldsymbol{v}(t)$ are learned first, followed by non-salient ones. **Right:** Entropy of the attention (i.e., entropy(softmax($\boldsymbol{v}^2$))) drops when salient components are learned first, and then rebounces when other components catch up.

**Theorem 4** (Convergence speed of salient vs. non-salient components). *Let $\delta_j(t) := 1 - v_j(t)/\mu_j$ be the convergence metric for component $j$ ($\delta_j(t) = 0$ means that the component $j$ converges). For nonlinear dynamics with attention (Eqn. 7), then*

$$\frac{\ln \delta_j(0)/\delta_j(t)}{\ln \delta_k(0)/\delta_k(t)} = \frac{e^{\mu_j^2/2}}{e^{\mu_k^2/2}}(1 + \Lambda_{jk}(t)) \tag{8}$$

*Here $\Lambda_{jk}(t) = \lambda_{jk}(t) \cdot e^{\mu_k^2/2} \ln^{-1}(\delta_k(0)/\delta_k(t))$ where $|\lambda_{jk}(t)| \le C_{jk}$ and $C_{jk}$ only depends on $\delta_j(0)$ and $\delta_k(0)$. So when $|\delta_k(t)| \ll |\delta_k(0)| \exp[-C_{jk} \exp(\mu_k^2)]$, we have $|\Lambda(t)| \ll 1$.*

**Remarks.** For linear attention, the ratio is different but the derivation is similar and simpler. Note that the convergence speed heavily depends on the magnitude of $\mu_j$. If $\mu_j > \mu_k$, then $\delta_j(t) \ll \delta_k(t)$ and $v_j(t)$ converges much faster than $v_k(t)$. Therefore, the salient (i.e., large) components is learned first, and the non-salient (i.e., small) component is learned later, due to the modulation of the extra term $\exp(\boldsymbol{v}^2/2)$ thanks to self-attention, as demonstrated in Fig. 4.

A follow-up question arises: What is the intuition behind salient and non-salient components in $\boldsymbol{\mu}$? Note that $\boldsymbol{\mu}$ is an $\ell_2$-normalized version of the conditional token frequency $\boldsymbol{x}$, given the query $q = m$. In this case, similar to Theorem 2 (and Tian et al. (2023)), we again see that if a contextual token $l$ co-occurs a lot with the query $m$, then the corresponding component $\mu_l$ becomes larger and the growth speed of $v_l$ towards $\mu_l$ is much faster.

**Relationship with rank of MLP lower layer**. Since MLP and attention layer has joint dynamics (Theorem 1), this also suggests that in the MLP layer, the rank of lower layer matrix $W$ (which projects into the hidden nodes) will first drop since the weight components that correspond to high target value $\mu_j$ grow first, and then bounce back to higher rank when the components that correspond to low target value $\mu_j$ catch up later.

## 5 How self-attention learns hierarchical data distribution?

A critical difference between the training dynamics of linear and nonlinear MLP is that in the nonlinear case, although slowly, the non-salient components will still grow, and the entropy of the attention bounces back later. While for 1-layer Transformer, this may only slow the training with no clear benefits, the importance of such a behavior is manifested if we think about the dynamics of multiple Transformer layers trained on a data distribution generated in a hierarchical manner.

Consider a simple generative hierarchical binary latent tree model (`HBLT`) (Tian et al., 2020) (Fig. 7(a)) in which we have latent (unobservable) binary variables $y$ at layer $s$ that generate latents at layer $s-1$, until the observable tokens are generated at the lowest level ($s = 0$). The topmost layer is the class label $y_0$, which can take $D$ discrete values. In `HBLT`, the generation process of $y_\beta$ at layer $s-1$ given $y_\alpha$ at layer $s$ can be characterized by their conditional probability $\mathbb{P}[y_\beta = 1|y_\alpha = 1] = \mathbb{P}[y_\beta = 0|y_\alpha = 0] = \frac{1}{2}(1 + \rho)$. The *uncertainty* hyperparameter $\rho \in [-1, 1]$ determines how much the top level latents can determine the values of the low level ones. Please check Appendix A.5 for its formal definition.

With `HBLT`, we can compute the co-occurrence frequency of two tokens $l$ and $m$, as a function of the depth of their common latent ancestor (CLA):

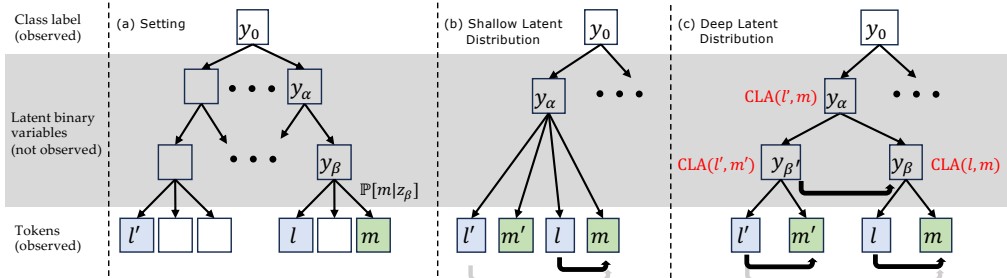

Figure 5: **(a)** Hierarchical binary tree generative models. Except for $y_0$ that is the observable label of a sequence and can take $D$ discrete labels, all latent variables follow binomial distribution. A binary leaf variable $y_l = 1$ indicates that token $l$ appears in the sequence. **(b)** Attention dynamics in multi-layer setting. There is a strong co-occurrence between the query $m$ and the token $l$, but a weak co-occurrence between $m$ and $l'$. As a result, $m$ associates with $l$ first, and eventually associates with $l'$, even if they co-occur weakly, according to Theorem 4. **(c)** If there exists an additional layer $y_\beta$ and $y_{\beta'}$ in the latent hierarchy, the association $m$-$l$ and $m'$-$l'$ will be learned first due to their high co-occurrence. Once the lower hierarchy gets learned and some hidden nodes in MLP represents $y_\beta$ and $y_{\beta'}$ (see Sec. 6 for experimental validation), on the next level, $y_\beta$ and $y_{\beta'}$ shows strong co-occurrence and gets picked up by the self-attention mechanism to form even higher level features. In contrast, the association of $l'$-$m$ is much slower and does not affect latent hierarchy learning, showing that self-attention mechanism is adaptive to the structure of data distribution.

**Theorem 5** (Token Co-occurrence in $\texttt{HBLT}(\rho)$). *If token $l$ and $m$ have common latent ancestor (CLA) of depth $H$ (Fig. 5(c)), then $\mathbb{P}[y_l = 1|y_m = 1] = \frac{1}{2}\left(\frac{1 + \rho^{2H} - 2\rho^{L-1}\rho_0}{1 - \rho^{L-1}\rho_0}\right)$, where $L$ is the total depth of the hierarchy and $\rho_0 := \boldsymbol{p}_{\cdot|0}^\top \boldsymbol{p}_0$, in which $\boldsymbol{p}_0 = [\mathbb{P}[y_0 = k]] \in \mathbb{R}^D$ and $\boldsymbol{p}_{\cdot|0} := [\mathbb{P}[y_l = 0|y_0 = k]] \in \mathbb{R}^D$, where $\{y_l\}$ are the immediate children of the root node $y_0$.*

**Remarks.** If $y_0$ takes multiple values (many classes) and each class only trigger one specific latent binary variables, then most of the top layer latents are very sparsely triggered and thus $\rho_0$ is very close to 1. If $\rho$ is also close to 1, then for deep hierarchy and shallow common ancestor, $\mathbb{P}[y_l = 1|y_m = 1] \to 1$. To see this, assume $\rho = \rho_0 = 1 - \epsilon$, then we have:

$$\mathbb{P}[y_l = 1|y_m = 1] = \frac{1}{2}\left[\frac{1 + 1 - 2H\epsilon - 2(1 - L\epsilon)}{1 - (1 - L\epsilon)}\right] + O(\epsilon^2) = 1 - \frac{H}{L} + O(\epsilon^2) \tag{9}$$

This means that two tokens $l$ and $m$ co-occur a lot, if they have a shallow CLA ($H$ small) that is close to both tokens. If their CLA is high in the hierarchy (e.g., $l'$ and $m$), then the token $l'$ and $m$ have much weaker co-occurrence and $\mathbb{P}(l'|m)$ (and thus $x_{l'}$ and $\mu_{l'}$) is small.

With this generative model, we can analyze qualitatively the learning dynamics of $\texttt{JoMA}$: first it focuses on associating the tokens in the same lowest hierarchy as the query $m$ (and these tokens co-occur a lot with $m$), then gradually reaches out to other tokens $l'$ that co-occur less with $m$, if they **have not been picked up** by other tokens (Fig. 5(b)); if $l'$ co-occurs a lot with some other $m'$, then $m$-$l$ and $m'$-$l'$ form their own lower hierarchy, respectively. This leads to learning of high-level features $y_\beta$ and $y_{\beta'}$, which has high correlation are associated in the higher level. Therefore, the latent hierarchy is implicitly learned.

## 6 EXPERIMENTS

**Dynamics of Attention Sparsity**. Fig. 6 shows how attention sparsity changes over time when training from scratch. We use $10^{-4}$ learning rate and test our hypothesis on Wikitext2/Wikitext103 (Merity et al., 2016) (top/bottom row). Fig. 8 further shows that different learning rate leads to different attention sparsity patterns. With large learning rate, attention becomes extremely sparse as in (Tian et al., 2023). Interestingly, the attention patterns, which coincide with our theoretical analysis, yield the best validation score.

We also tested our hypothesis in OPT (Zhang et al., 2022) (OPT-2.7B) and Pythia (Biderman et al., 2023) (Pythia-70M/1.4B/6.9B) pre-trained models, both of which has public intermediate checkpoints. While the attention patterns show less salient drop-and-bounce patterns, the dynamics of stable ranks of the MLP lower layer (projection into hidden neurons) show much salient such structures for top layers, and dropping curves for bottom layers since they are suppressed by top-level learning (Sec. 5). Note that stable ranks only depend on the model parameters and thus may be more reliable than attention sparsity.

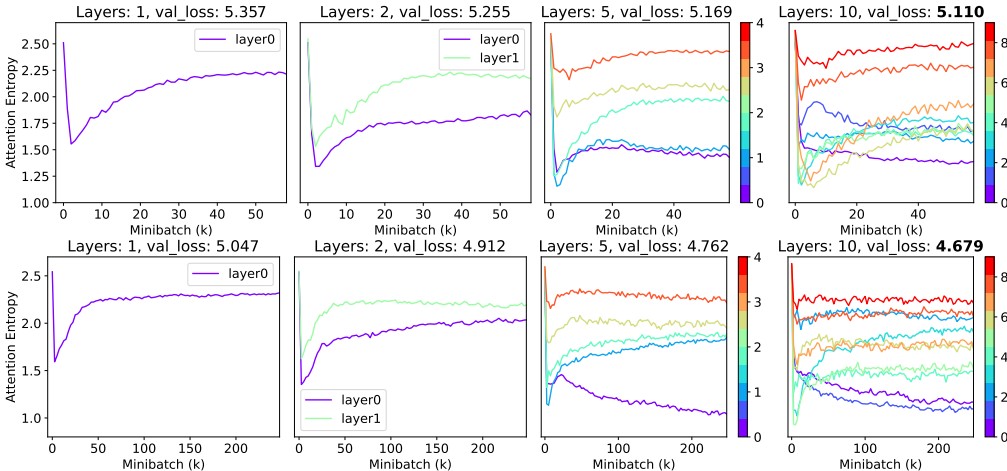

Figure 6: Dynamics of attention sparsity. In 1-layer setting, The curves bear strong resemblance to our theoretical prediction (Fig. 4); in multi-layer settings, the attention entropy in top Transformer layers has a similar shape, while the entropy in bottom layers are suppressed due to layer interactions (Sec. 4). **Top row:** Wikitext2, **Bottom row:** Wikitext103.

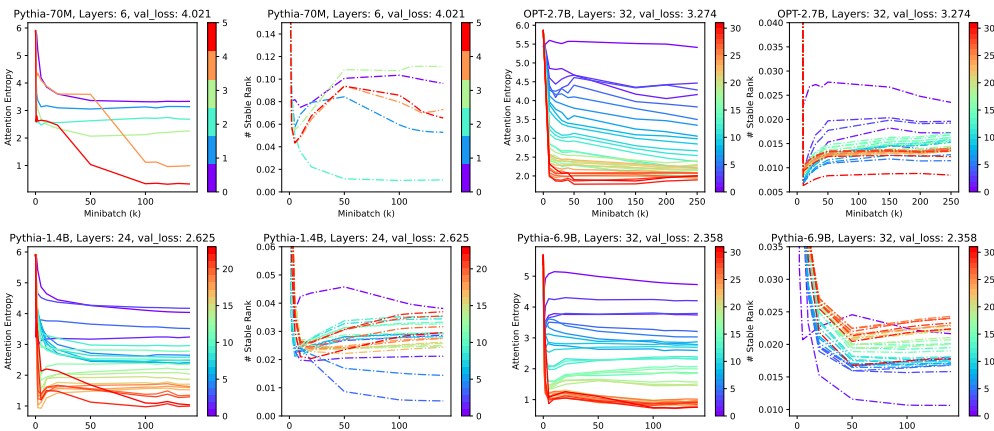

Figure 7: Dynamics of attention sparsity and stable rank in OPT-2.7B and Pythia-70M/1.4B/6.9B. Results are evaluated on Wikitext103 (Merity et al., 2016).

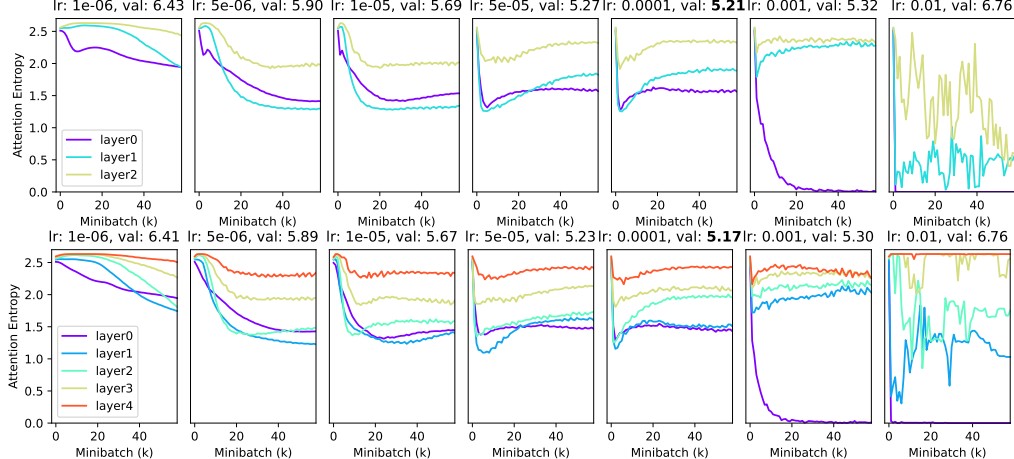

Figure 8: Effect of different learning rates on attention sparsity. Different learning rates lead to different dynamics of attention sparsity, and the attention patterns consistent with our theoretical analysis (Fig. 4) give the lowest validation losses.

| $(N_0, N_1)$ | $C = 20, N_{ch} = 2$ | | $C = 20, N_{ch} = 3$ | | $C = 30, N_{ch} = 2$ | |
|---|---|---|---|---|---|---|
| | (10, 20) | (20, 30) | (10, 20) | (20, 30) | (10, 20) | (20, 30) |
| NCorr $(s = 0)$ | $0.99 \pm 0.01$ | $0.97 \pm 0.02$ | $1.00 \pm 0.00$ | $0.96 \pm 0.02$ | $0.99 \pm 0.01$ | $0.94 \pm 0.04$ |
| NCorr $(s = 1)$ | $0.81 \pm 0.05$ | $0.80 \pm 0.05$ | $0.69 \pm 0.05$ | $0.68 \pm 0.04$ | $0.73 \pm 0.08$ | $0.74 \pm 0.03$ |
| $(N_0, N_1)$ | $C = 30\ N_{ch} = 3$ | | $C = 50, N_{ch} = 2$ | | $C = 50, N_{ch} = 3$ | |
| | (10, 20) | (20, 30) | (10, 20) | (20, 30) | (10, 20) | (20, 30) |
| NCorr $(s = 0)$ | $0.99 \pm 0.01$ | $0.95 \pm 0.03$ | $0.99 \pm 0.01$ | $0.95 \pm 0.03$ | $0.99 \pm 0.01$ | $0.95 \pm 0.03$ |
| NCorr $(s = 1)$ | $0.72 \pm 0.04$ | $0.66 \pm 0.02$ | $0.58 \pm 0.02$ | $0.55 \pm 0.01$ | $0.64 \pm 0.02$ | $0.61 \pm 0.04$ |

Table 1: Normalized correlation between the latents and their best matched hidden node in MLP of the same layer. All experiments are run with 5 random seeds.

**Validation of Alignment between latents and hidden nodes in MLP**. Sec. 5 is based on an assumption that the hidden nodes in MLP layer will learn the latent variables. We verify this assumption in synthetic data sampled by `HBLT`, which generate latent variables in a top-down manner, until the final tokens are generated. The latent hierarchy has 2 hyperparameters: number of latents per layer ($N_s$) and number of children per latent ($N_{ch}$). $C$ is the number of classes. Adam optimizer is used with learning rate $10^{-5}$. Vocabulary size $M = 100$, sequence length $T = 30$ and embedding dimension $d = 1024$.

We use 3-layer generative model as well as 3-layer Transformer models. We indeed perceive high correlations between the latents and the hidden neurons between corresponding layers. Note that latents are known during input generation procedure but are not known to the transformer being trained. We take the maximal activation of each neuron across the sequence length, and compute normalized correlation between maximal activation of each neuron and latents, after centeralizing across the sample dimension. Tbl. 1 shows that indeed in the learned models, for each latent, there exists at least one hidden node in MLP that has high normalized correlation with it, in particular in the lowest layer. When the generative models becomes more complicated (i.e., both $N_{ch}$ and $N_l$ become larger), the correlation goes down a bit.

## 7 Discussion

**Deal with almost orthogonal embeddings**. In this paper, we focus on *fixed* orthonormal embeddings vectors. However, in real-world Transformer training, the assumption may not be valid, since often the embedding dimension $d$ is smaller than the number of vocabulary $M$ so the embedding vectors cannot be orthogonal to each other. In this setting, one reasonable assumption is that the embedding vectors are *almost* orthogonal. Thanks to Johnson–Lindenstrauss lemma, one interesting property of high-dimensional space is that for $M$ embedding vectors to achieve almost orthogonality $|\boldsymbol{u}_l^\top \boldsymbol{u}_{l'}| \leq \epsilon$, only $d \geq 8\epsilon^{-2} \log M$ is needed. As a result, our `JoMA` framework (Theorem 1) will have additional $\epsilon$-related terms and we leave the detailed analysis as one of our future work.

**Training embedding vectors**. Another factor that is not considered in `JoMA` is that the embedding vectors are also trained simultaneously. This could further boost the efficiency of Transformer architecture, since concepts with similar semantics will learn similar embeddings. This essentially reduces the vocabulary size at each layer for learning to be more effective, and leads to better generalization. For example, in each hidden layer $4d$ hidden neurons are computed, which does not mean there are $4d$ independent intermediate "tokens", because many of their embeddings are highly correlated.

**Self-attention computed from embedding**. `JoMA` arrives at the joint dynamics of MLP and attention by assuming that the pairwise attention score $Z$ is an independent parameters optimized under SGD dynamics. In practice, $Z = UW_Q W_K^\top U^\top$ is also parameterized by the embedding matrix, which allow generalization to tokens with similar embeddings, and may accelerate the training dynamics of $Z$. We leave it in the future works.

## 8 Conclusion

We propose `JoMA`, a framework that characterizes the joint training dynamics of nonlinear MLP and attention layer, by integrating out the self-attention logits. The resulting dynamics connects the dynamics of nonlinear MLP lower layer weights (projection into hidden neurons) and self-attention, and shows that the attention first becomes sparse (or weights becomes low rank) and then becomes dense (or weights becomes high rank). Furthermore, we qualitatively give a learning mechanism of multilayer Transformer that reveals how self-attentions at different layers interact with each other to learn the latent feature hierarchy.

ACKNOWLEDGMENTS

Simon S. Du is supported by supported by NSF IIS 2110170, NSF DMS 2134106, NSF CCF 2212261, NSF IIS 2143493, NSF CCF 2019844, NSF IIS 2229881.

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

## A    Proofs

### A.1    Per-hidden loss formulation

Our Assumption 1 has an equivalent per-hidden node loss:

$$\max_{\{\boldsymbol{w}_k\},\{\boldsymbol{z}_m\}} \mathbb{E}_{\mathcal{D}}\left[\sum_k g_{h_k} h_k\right] := \max_{\{\boldsymbol{w}_k\},\{\boldsymbol{z}_m\}} \mathbb{E}_{i \sim \mathcal{D}}\left[\sum_k g_{h_k}[i] h_k[i]\right] \tag{10}$$

where $g_{h_k}[i]$ is the backpropagated gradient sent to node $h_k$ at sample $i$.

### A.2    JoMA framework (Section 3)

**Theorem 1** (JoMA). *Let $\boldsymbol{v}_k := U_C^\top \boldsymbol{w}_k$, then the dynamics of Eqn. 2 satisfies the invariants:*

- *Linear attention. The dynamics satisfies $\boldsymbol{z}_m^2(t) = \sum_k \boldsymbol{v}_k^2(t) + \boldsymbol{c}$.*

- *Exp attention. The dynamics satisfies $\boldsymbol{z}_m(t) = \frac{1}{2}\sum_k \boldsymbol{v}_k^2(t) + \boldsymbol{c}$.*

- *Softmax attention. If $\bar{\boldsymbol{b}}_m := \mathbb{E}_{q=m}[\boldsymbol{b}]$ is a constant over time and $\overline{\mathbb{E}_{q=m}\left[\sum_k g_{h_k} h_k' \boldsymbol{b}\boldsymbol{b}^\top\right]} = \bar{\boldsymbol{b}}_m \mathbb{E}_{q=m}[\sum_k g_{h_k} h_k' \boldsymbol{b}]$, then the dynamics satisfies $\boldsymbol{z}_m(t) = \frac{1}{2}\sum_k \boldsymbol{v}_k^2(t) - \|\boldsymbol{v}_k(t)\|_2^2 \bar{\boldsymbol{b}}_m + \boldsymbol{c}$.*

*Under zero initialization ($\boldsymbol{w}_k(0) = 0$, $\boldsymbol{z}_m(0) = 0$), then the time-independent constant $\boldsymbol{c} = 0$.*

*Proof.* Let $L := \partial \boldsymbol{b}/\partial \boldsymbol{z}_m$. Plugging the dynamics of $\boldsymbol{w}_k$ into the dynamics of self-attention logits $\boldsymbol{z}_m$, we have:

$$\dot{\boldsymbol{z}}_m = \mathbb{E}_{q=m}\left[L^\top U_C^\top \sum_k g_{h_k} h_k' \boldsymbol{w}_k\right] = \sum_k \mathbb{E}_{q=m}\left[g_{h_k} h_k' L^\top \boldsymbol{v}_k\right] \tag{11}$$

Before we start, we first define $\xi_k(t) := \int_0^t \mathbb{E}_{q=m}[g_{h_k}(t')h_k'(t')]\,\mathrm{d}t'$. Therefore, $\dot{\xi}_k = \mathbb{E}_{q=m}[g_{h_k}h_k']$. Intuitively, $\xi_k$ is the bias of node $k$, regardless of whether there exists an actual bias parameter to optimize.

Notice that $U_C^\top \boldsymbol{f} = \boldsymbol{b} + U_C^\top \boldsymbol{u}_q$, with orthonormal condition between contextual and query tokens: $U_C^\top \boldsymbol{u}_m = 0$, and thus $U_C^\top \boldsymbol{f} = \boldsymbol{b}$, which leads to

$$\dot{\boldsymbol{v}}_k = U_C^\top \dot{\boldsymbol{w}}_k = U_C^\top \mathbb{E}_{q=m}[g_{h_k}h_k' \boldsymbol{f}] = \mathbb{E}_{q=m}[g_{h_k}h_k' \boldsymbol{b}] \tag{12}$$

**Unnormalized attention** ($A := \text{const}$). In this case, we have $\boldsymbol{b} = \sigma(\boldsymbol{z}_m) \circ \boldsymbol{x}/A$ and $L = \text{diag}(\sigma'(\boldsymbol{z}_m) \circ \boldsymbol{x})/A = \text{diag}\left(\frac{\sigma'(\boldsymbol{z}_m)}{\sigma(\boldsymbol{z}_m)}\right)\text{diag}(\boldsymbol{b})$ and thus

$$\dot{\boldsymbol{z}}_m = \sum_k \mathbb{E}_{q=m}[g_{h_k}h_k' L^\top \boldsymbol{v}_k] = \text{diag}\left(\frac{\sigma'(\boldsymbol{z}_m)}{\sigma(\boldsymbol{z}_m)}\right)\sum_k \mathbb{E}_{q=m}[g_{h_k}h_k' \boldsymbol{b}] \circ \boldsymbol{v}_k \tag{13}$$

$$= \text{diag}\left(\frac{\sigma'(\boldsymbol{z}_m)}{\sigma(\boldsymbol{z}_m)}\right)\sum_k \dot{\boldsymbol{v}}_k \circ \boldsymbol{v}_k \tag{14}$$

which leads to

$$\text{diag}\left(\frac{\sigma(\boldsymbol{z}_m)}{\sigma'(\boldsymbol{z}_m)}\right)\dot{\boldsymbol{z}}_m = \sum_k \dot{\boldsymbol{v}}_k \circ \boldsymbol{v}_k \tag{15}$$

Therefore, for linear attention, $\sigma(\boldsymbol{z}_m)/\sigma'(\boldsymbol{z}_m) = \boldsymbol{z}_m$, by integrating both sides, we have $\boldsymbol{z}_m^2(t) = \sum_k \boldsymbol{v}_k^2(t) + \boldsymbol{c}$. For exp attention, $\sigma(\boldsymbol{z}_m)/\sigma'(\boldsymbol{z}_m) = 1$, then by integrating both sides, we have $\boldsymbol{z}_m(t) = \frac{1}{2}\sum_k \boldsymbol{v}_k^2(t) + \boldsymbol{c}$.

**Softmax attention.** In this case, we have $L = \text{diag}(\boldsymbol{b}) - \boldsymbol{b}\boldsymbol{b}^\top$. Therefore,

$$\mathbb{E}_{q=m}[g_{h_k}h_k'\text{diag}(\boldsymbol{b})]U_C^\top \boldsymbol{w}_k = \mathbb{E}_{q=m}[g_{h_k}h_k' \boldsymbol{b}] \circ \boldsymbol{v}_k = \dot{\boldsymbol{v}}_k \circ \boldsymbol{v}_k \tag{16}$$

where $\circ$ is the Hadamard (element-wise) product. Now Therefore, we have:

$$\mathbb{E}_{q=m}\left[g_{h_k}h_k'\boldsymbol{b}^\top\right]U_C^\top\boldsymbol{w}_k = \dot{\boldsymbol{v}}_k^\top\boldsymbol{v}_k \tag{17}$$

Given the assumption that $\boldsymbol{b}$ is uncorrelated with $\sum_k g_{h_k}h_k'\boldsymbol{b}$ (e.g., due to top-down gradient information), and let $\bar{\boldsymbol{b}}_m = \mathbb{E}_{q=m}\left[\boldsymbol{b}\right]$, we have:

$$\dot{\boldsymbol{z}}_m = \sum_k \dot{\boldsymbol{v}}_k \circ \boldsymbol{v}_k - \bar{\boldsymbol{b}}_m \dot{\boldsymbol{v}}_k^\top\boldsymbol{v}_k \tag{18}$$

If we further assume that $\bar{\boldsymbol{b}}_m$ is constant over time, then we can integrate both side to get a close-form solution between $\boldsymbol{z}_m(t)$ and $\{\boldsymbol{v}_k(t)\}$:

$$\boldsymbol{z}_m(t) = \frac{1}{2}\sum_k \left(\boldsymbol{v}_k^2 - \|\boldsymbol{v}_k\|_2^2\bar{\boldsymbol{b}}_m\right) + \boldsymbol{c} \tag{19}$$

$\square$

**Theorem 2** (Linear Dynamics with Self-attention). *With linear MLP activation and zero initialization, for exp attention any two tokens $l \neq l'$ satisfy the following invariants:*

$$\frac{\mathrm{erf}\left(v_l(t)/2\right)}{\Delta_{lm}} = \frac{\mathrm{erf}(v_{l'}(t)/2)}{\Delta_{l'm}} \tag{4}$$

*where $\Delta_{lm} = \mathbb{E}_{q=m}\left[g_{h_k}x_l\right]$ and $\mathrm{erf}(x) = \frac{2}{\sqrt{\pi}}\int_0^x e^{-t^2}\mathrm{d}t$ is Gauss error function.*

*Proof.* Due to the assumption, we have:

$$\dot{v}_l = \mathbb{E}_{q=m}\left[g_{h_k}x_l\right]\exp(z_{ml})/A = \Delta_{lm}\exp(z_{ml})/A \tag{20}$$

where $\Delta_{lm} := \mathbb{E}_{q=m}\left[g_{h_k}x_l\right]$. If $x_l[i] = \mathbb{P}(l|m, y[i])$, then $\Delta_{lm} = \mathbb{E}_{l,q=m}\left[g_{h_k}\right]\mathbb{P}(l|m)$. Note that for linear model, $\Delta_{lm}$ is a constant over time.

Plugging in the close-form solution for exp attention, the dynamics becomes

$$\dot{v}_l = \Delta_{lm}\exp(v_l^2/2 + c_l)/A \tag{21}$$

Assuming $c_l = 0$, then for any two tokens $l \neq l'$, we get

$$\frac{\dot{v}_l}{\dot{v}_{l'}} = \frac{\Delta_{lm}\exp(z_{ml})}{\Delta_{l'm}\exp(z_{ml'})} = \frac{\Delta_{lm}\exp(v_l^2/2)}{\Delta_{l'm}\exp(v_{l'}^2/2)} \tag{22}$$

which can be integrated using $\mathrm{erf}(\cdot)$ function (i.e., Gaussian CRF: $\mathrm{erf}(x) = \frac{2}{\sqrt{\pi}}\int_0^x e^{-t^2}\mathrm{d}t$):

$$\frac{\mathrm{erf}\left(v_l(t)/2\right)}{\Delta_{lm}} = \frac{\mathrm{erf}(v_{l'}(t)/2)}{\Delta_{l'm}} + c_{ll'} \tag{23}$$

if $\boldsymbol{v}(0) = 0$, then $c_{ll'} = 0$. $\square$

### A.3 Dynamics of Nonlinear activations (Sec. 4)

#### A.3.1 Without self-attention (or equivalently, with uniform attention)

**Lemma 1** (Expectation of Hyperplane function under Isotropic distribution). *For any isotropic distribution $p(\boldsymbol{x} - \bar{\boldsymbol{x}})$ with mean $\bar{\boldsymbol{x}}$ in a subspace spanned by orthonormal bases $R$, if $\boldsymbol{v} \neq \boldsymbol{0}$, we have:*

$$\mathbb{E}_p\left[\boldsymbol{x}\psi(\boldsymbol{v}^\top\boldsymbol{x} + \xi)\right] = \frac{\theta_1(r_{\boldsymbol{v}})}{\|\boldsymbol{v}\|_2}\bar{\boldsymbol{x}} + \frac{\theta_2(r_{\boldsymbol{v}})}{\|\boldsymbol{v}\|_2^3}RR^\top\boldsymbol{v}, \qquad \mathbb{E}_p\left[\psi(\boldsymbol{v}^\top\boldsymbol{x} + \xi)\right] = \frac{\theta_1(r_{\boldsymbol{v}})}{\|\boldsymbol{v}\|_2} \tag{24}$$

*where $r_{\boldsymbol{v}} := \boldsymbol{v}^\top\bar{\boldsymbol{x}} + \xi$ is the (signed) distance between the distribution mean $\bar{\boldsymbol{x}}$ and the affine hyperplane $(\boldsymbol{v}, \xi)$. $\theta_1(r)$ and $\theta_2(r)$ only depends on $\psi$ and the underlying distribution but not $\boldsymbol{v}$. Additionally,*

- *If $\psi(r)$ is monotonously increasing, then $\theta_1(r)$ is also monotonous increasing;*

- If $\psi(r) \geq 0$, then $\theta_1(r) \geq 0$;

- If $\psi(-\infty) = 0$, $\psi(+\infty) = 1$, then $\theta_1(-\infty) = 0$ and $\theta_1(+\infty) = 1$;

- If $\psi(-\infty) = 0$, then $\theta_2(-\infty) = 0$.

*Proof.* Note that $\boldsymbol{x}'$ is isotropic in $\mathrm{span}(R)$ and thus $p(\boldsymbol{x}')$ just depends on $\|\boldsymbol{x}'\|$, we let $p_0 : \mathbb{R}^+ \to \mathbb{R}^+$ satisfies $p_0(\|\boldsymbol{x}'\|) = p(\boldsymbol{x}')$. Our goal is to calculate

$$\mathbb{E}_p\left[\boldsymbol{x}\psi(\boldsymbol{w}^\top\boldsymbol{x} + \xi)\right] = \int_{\mathrm{span}(R)} \boldsymbol{x}\psi(\boldsymbol{w}^\top\boldsymbol{x} + \xi)p(\boldsymbol{x} - \boldsymbol{\mu})\mathrm{d}\boldsymbol{x} \tag{25}$$

$$= \int_{\mathrm{span}(R)} (\boldsymbol{x}' + \boldsymbol{\mu})\psi(\boldsymbol{w}^\top\boldsymbol{x}' + r_{\boldsymbol{w}})p(\boldsymbol{x}')\mathrm{d}\boldsymbol{x}' \tag{26}$$

where $\boldsymbol{x}' := \boldsymbol{x} - \boldsymbol{\mu}$ is isotropic. Since $RR^\top\boldsymbol{w}$ is the projection of $\boldsymbol{w}$ onto space $\mathrm{span}(R)$, we denote $\boldsymbol{v} := RR^\top\boldsymbol{w}$ and $y' := \boldsymbol{w}^\top\boldsymbol{x}' = \boldsymbol{v}^\top\boldsymbol{x}'$ since $\boldsymbol{x}'$ lies in $\mathrm{span}(R)$. Then let $S$ be any hyper-plane through $\boldsymbol{v}$, which divide $\mathrm{span}(R)$ into two symmetric part $V_+$ and $V_-$(Boundary is zero measurement set and can be ignored), we have,

$$P_1 := \int_{\mathrm{span}(R)} \boldsymbol{x}'\psi(\boldsymbol{w}^\top\boldsymbol{x}' + r_{\boldsymbol{w}})p(\boldsymbol{x}')\mathrm{d}\boldsymbol{x}' \tag{27}$$

$$= (\int_{V_+} + \int_{V_-})\boldsymbol{x}'\psi(\boldsymbol{v}^\top\boldsymbol{x}' + r_{\boldsymbol{w}})p(\boldsymbol{x}')\mathrm{d}\boldsymbol{x}' \tag{28}$$

$$= 2 \times \int_{V_+} \frac{\boldsymbol{v}^\top\boldsymbol{x}'}{\|\boldsymbol{v}\|} \cdot \frac{\boldsymbol{v}}{\|\boldsymbol{v}\|} \cdot \psi(\boldsymbol{v}^\top\boldsymbol{x}' + r_{\boldsymbol{w}})p(\boldsymbol{x}')\mathrm{d}\boldsymbol{x}' \tag{29}$$

$$= \{\int_{\mathrm{span}(R)} y'\psi(y' + r_{\boldsymbol{w}})p(\boldsymbol{x}')\mathrm{d}\boldsymbol{x}'\} \cdot \frac{\boldsymbol{v}}{\|\boldsymbol{v}\|^2} \tag{30}$$

Eqn. 29 holds since for every $\boldsymbol{x}' \in V_+$, we can always find unique $\boldsymbol{x}'' \in V_-$ defined as

$$\boldsymbol{x}'' = -(\boldsymbol{x}' - \frac{\boldsymbol{v}^\top\boldsymbol{x}'}{\|\boldsymbol{v}\|^2}\boldsymbol{v}) + \frac{\boldsymbol{v}^\top\boldsymbol{x}'}{\|\boldsymbol{v}\|^2}\boldsymbol{v} = \frac{2y'}{\|\boldsymbol{v}\|^2}\boldsymbol{v} - \boldsymbol{x}' \tag{31}$$

where $\boldsymbol{x}''$ and $\boldsymbol{x}'$ satisfy $\|\boldsymbol{x}''\| = \|\boldsymbol{x}'\|$, $\boldsymbol{v}^\top\boldsymbol{x}'' = \boldsymbol{v}^\top\boldsymbol{x}'$, and have equal reverse component $\pm(\boldsymbol{x}' - \frac{\boldsymbol{v}^\top\boldsymbol{x}'}{\|\boldsymbol{v}\|^2}\boldsymbol{v})$ perpendicular to $\boldsymbol{v}$. Thus for the $\boldsymbol{x}'$ in Eqn. 28, only the component parallel to $\boldsymbol{v}$ remains. Furthermore, let $\{\boldsymbol{u}_1, \ldots, \boldsymbol{u}_{n-1}, \boldsymbol{v}/\|\boldsymbol{v}\|\}$ to be an orthonormal bases of $\mathrm{span}(R)$ and denote $x_i' := \boldsymbol{u}_i^\top\boldsymbol{x}', \forall i \in [n-1]$, then we have

$$P_1 = \{\int_{y'} y'\psi(y' + r_{\boldsymbol{w}})\mathrm{d}(\frac{y'}{\|\boldsymbol{v}\|})[\int_{x_1'} \cdots \int_{x_{n-1}'} p(\boldsymbol{x}')\mathrm{d}x_1' \ldots \mathrm{d}x_{n-1}']\} \cdot \frac{\boldsymbol{v}}{\|\boldsymbol{v}\|^2} \tag{32}$$

$$=: \{\int_{-\infty}^{+\infty} y'\psi(y' + r_{\boldsymbol{w}})p_n(y')\mathrm{d}y'\} \cdot \frac{\boldsymbol{v}}{\|\boldsymbol{v}\|^3} \tag{33}$$

Here $p_n(y')$ is the probability density function of $y'$ obtained from $\boldsymbol{x}'$. For the trivial case where $n = 1$, clearly $p_n(y') = p_0(|y'|) = p(y')$. If $n \geq 2$, it can be further calculated as:

$$p_n(y') = \int_{x_1'} \cdots \int_{x_{n-1}'} p_0(\sqrt{(x_1')^2 + \ldots + (x_{n-1}')^2 + (y')^2}) \cdot \mathrm{d}x_1' \ldots \mathrm{d}x_{n-1} \tag{34}$$

$$= \int_0^{+\infty} p_0(\sqrt{y'^2 + l^2}) \cdot S_{n-1}(l)\mathrm{d}l \tag{35}$$

$$= \frac{(n-1)\pi^{(n-1)/2}}{\Gamma(\frac{n+1}{2})} \int_0^{+\infty} p_0(\sqrt{y'^2 + l^2}) \cdot l^{n-2}\mathrm{d}l \tag{36}$$

$$= \begin{cases} \frac{2^{n/2}\pi^{n/2-1}}{(n-3)!!} \int_0^{+\infty} p_0(\sqrt{y'^2 + l^2}) \cdot l^{n-2}\mathrm{d}l, & n \text{ is even} \\ \frac{2\pi^{(n-1)/2}}{(\frac{n-3}{2})!} \int_0^{+\infty} p_0(\sqrt{y'^2 + l^2}) \cdot l^{n-2}\mathrm{d}l, & n \text{ is odd} \end{cases} \tag{37}$$

where $S_n(R) = \frac{n\pi^{n/2}}{\Gamma(n/2+1)} R^{n-1}$ represents the surface area of an $n$-dimensional hyper-sphere of radius $l$. $\Gamma$ denotes the gamma function and we use the property that $\Gamma(n+1) = n!$ and $\Gamma(n + \frac{1}{2}) = (2n-1)!!\sqrt{\pi}2^{-n}$ for any $n \in \mathbb{N}^+$.

Similarly, for another term we have

$$P_2 = \int_{\text{span}(R)} \boldsymbol{\mu} \cdot \psi(\boldsymbol{w}^\top \boldsymbol{x}' + r_{\boldsymbol{w}}) p(\boldsymbol{x}') \mathrm{d}\boldsymbol{x}' \tag{38}$$

$$= \{\int_{-\infty}^{+\infty} \psi(y' + r_{\boldsymbol{w}}) p_n(y') \mathrm{d}y'\} \cdot \frac{\boldsymbol{\mu}}{\|\boldsymbol{v}\|} \tag{39}$$

$$\tag{40}$$

Finally, let

$$\theta_1(r_{\boldsymbol{w}}) := \int_{-\infty}^{+\infty} \psi(y' + r_{\boldsymbol{w}}) p_n(y') \mathrm{d}y' \tag{41}$$

$$\theta_2(r_{\boldsymbol{w}}) := \int_{-\infty}^{+\infty} y' \cdot \psi(y' + r_{\boldsymbol{w}}) p_n(y') \mathrm{d}y' \tag{42}$$

Then we arrive at the conclusion. $\qquad\square$

**Theorem 3** (Dynamics of nonlinear activation with uniform attention). *If $\boldsymbol{x}$ is sampled from a mixture of $C$ isotropic distributions centered at $[\bar{\boldsymbol{x}}_1, \ldots, \bar{\boldsymbol{x}}_C]$, where each $\bar{\boldsymbol{x}}_c \in \mathbb{R}^d$ and gradient $g_{h_k}$ are constant within each mixture, then:*

$$\dot{\boldsymbol{v}} = \Delta_m = \frac{1}{\|\boldsymbol{v}\|_2} \sum_c a_c \theta_1(r_c) \bar{\boldsymbol{x}}_c + \frac{1}{\|\boldsymbol{v}\|_2^3} \sum_c a_c \theta_2(r_c) \boldsymbol{v} \tag{5}$$

*here $a_c := \mathbb{E}_{q=m,c}[g_{h_k}]\mathbb{P}[c]$, $r_c := \boldsymbol{v}^\top \bar{\boldsymbol{x}}_c + \xi$ is the affinity to $\bar{\boldsymbol{x}}_c$ and the "bias" term $\xi(t) := \int_0^t \mathbb{E}_{q=m}[g_{h_k} h_k'] \mathrm{d}t$, $\theta_1$ and $\theta_2$ depend on derivative of nonlinearity $\psi := \phi'$ and data distribution but not $\boldsymbol{v}$. If $\psi$ is monotonous with $\psi(-\infty) = 0$ and $\psi(+\infty) = 1$, so does $\theta_1$.*

*Proof.* Since backpropagated gradient $g_{h_k}$ is constant within each of its mixed components, we have:

$$\Delta_m := \mathbb{E}_{q=m}[g_{h_k} h_k' \boldsymbol{b}] = \sum_j \mathbb{E}_{q=m,c=j}[g_{h_k} h_k' \boldsymbol{b}]\mathbb{P}[c = j] \tag{43}$$

$$= \sum_j \mathbb{E}_{q=m,c=j}[g_{h_k}]\mathbb{P}[c = j]\mathbb{E}_{q=m,c=j}[h_k' \boldsymbol{b}] \tag{44}$$

$$= \sum_j a_j \mathbb{E}_{\boldsymbol{x} \sim p(\boldsymbol{x} - \bar{\boldsymbol{x}}_j)}[\boldsymbol{b}\phi'(\boldsymbol{w}^\top \boldsymbol{f})] \tag{45}$$

Let $\psi = \phi'$. Note that $\boldsymbol{w}^\top \boldsymbol{f} = \boldsymbol{w}^\top(U_c \boldsymbol{b} + \boldsymbol{u}_q) = \boldsymbol{v}^\top \boldsymbol{b} + \xi$ and with uniform attention $\boldsymbol{b} = \boldsymbol{x}$, we have:

$$\Delta_m = \sum_j a_j \mathbb{E}_{\boldsymbol{x} \sim p(\boldsymbol{x} - \bar{\boldsymbol{x}}_j)}[\boldsymbol{x}\psi(\boldsymbol{v}^\top \boldsymbol{x} + \xi)] \tag{46}$$

Using Lemma 1 leads to the conclusion. $\qquad\square$

**Remarks**. Note that if $\phi$ is linear, then $\psi \equiv 1$, $\theta_1 \equiv 1$ and $\theta_2 \equiv 0$. In this case, $\theta_1$ is a constant, which marks a key difference between linear and nonlinear dynamics.

### A.3.2 (Tentative) Critical Point Analysis of Dynamics in Theorem 3

**Lemma 2** (Property of $\theta_1, \theta_2$ with homogeneous activation). *If $\phi(x) = x\phi'(x)$ is a homogeneous activation function and $\psi = \phi'$, then we have:*

$$\frac{\mathrm{d}}{\mathrm{d}r}(\theta_2(r) + r\theta_1(r)) = \theta_1(r) \tag{47}$$

*Integrating both sides and we get:*

$$\theta_2(r) + r\theta_1(r) = F(r) := \int_0^r \theta_1(r')\mathrm{d}r' + C \tag{48}$$

*Let $r = 0$ and it is clear that $C = \theta_2(0)$. Thus*

$$\theta_2(r) + r\theta_1(r) = F(r) = \int_0^r \theta_1(r')\mathrm{d}r' + \theta_2(0) \tag{49}$$

*If $\psi \geq 0$, then $F(r)$ is a monotonous increasing function with $F(+\infty) = +\infty$. Furthermore, if $\lim_{r \to -\infty} r\theta_1(r) = 0$ and $\psi(-\infty) = 0$, then $\theta_2(-\infty) = 0$ and $F(-\infty) = 0$ and thus $F(r) \geq 0$.*

*Proof.* Simply verify Eqn. 47 is true. $\square$

Overall, the dynamics can be quite complicated. We consider a special $C = 2$ case with one positive ($a_+$, $r_+$ and $\bar{\boldsymbol{x}}_+$) and one negative ($a_-$, $r_-$ and $\bar{\boldsymbol{x}}_-$) distribution.

**Lemma 3** (Existence of critical point of dynamics with ReLU activation). *For any homogeneous activation $\phi(x) = x\phi'(x)$, any stationary point of Eqn. 5 must satisfy $\sum_j a_j F(r_j) = 0$, where $F(r) := \theta_2(0) + \int_0^r \theta_1(r')\mathrm{d}r'$ is a monotonous increasing function.*

*Proof.* We rewrite the dynamics equations for the nonlinear activation without attention case:

$$\dot{\boldsymbol{v}} = \frac{1}{\|\boldsymbol{v}\|_2} \sum_j a_j \theta_1(r_j)\bar{\boldsymbol{x}}_j + \frac{1}{\|\boldsymbol{v}\|_2^3} \sum_j a_j \theta_2(r_j)\boldsymbol{v}, \qquad \dot{\xi} = \frac{1}{\|\boldsymbol{v}\|_2} \sum_j a_j \theta_1(r_j) \tag{50}$$

Notice that $\bar{\boldsymbol{x}}_j^\top \boldsymbol{v} = r_j - \xi$, this gives that:

$$\|\boldsymbol{v}\|_2 \boldsymbol{v}^\top \dot{\boldsymbol{v}} = \sum_j a_j \theta_1(r_j)(r_j - \xi) + \sum_j a_j \theta_2(r_j) \tag{51}$$

$$= \sum_j a_j(r_j \theta_1(r_j) + \theta_2(r_j)) - \xi \sum_j a_j \theta_1(r_j) \tag{52}$$

$$= \sum_j a_j F(r_j) - \|\boldsymbol{v}\|_2 \xi \dot{\xi} \tag{53}$$

in which the last equality is because the dynamics of $\xi$, and due to Lemma 2. Now we leverage the condition of stationary points ($\dot{\boldsymbol{v}} = 0$ and $\dot{\xi} = 0$), we arrive at the necessary conditions at the stationary points:

$$\sum_j a_j F(r_j) = 0 \tag{54}$$

Note that in general, the scalar condition above is only necessary but not sufficient. Eqn. 50 has $M_c + 1$ equations but we only have two scalar equations (Eqn. 50 and $\|\boldsymbol{v}\|_2 \dot{\xi} = \sum_j a_j \theta_1(r_j) = 0$). However, we can get a better characterization of the stationary points if there are only two components $a_+$ and $a_-$:

**A special case: one positive and one negative samples** In this case, we have (here $r_+ := \boldsymbol{v}^\top \bar{\boldsymbol{x}}_+ + \xi$ and $r_- := \boldsymbol{v}^\top \bar{\boldsymbol{x}}_- + \xi$):

$$a_+ F(r_+) - a_- F(r_-) = 0 \tag{55}$$

So the sufficient and necessary condition for $(\boldsymbol{v}, \xi)$ to be the critical point is that

$$\frac{F(r_+)}{F(r_-)} = \frac{\theta_1(r_+)}{\theta_1(r_-)} = \frac{a_-}{a_+} \tag{56}$$

Without loss of generality, we consider the case where $\phi$ is ReLU and $\psi(r) = \mathbf{I}[r > 0]$. Note that $\theta_1$ is a monotonously increasing function, we have $\theta_1^{-1} : (0, 1) \to \mathbb{R}$ such that $\theta_1^{-1}(\theta_1(r)) = r$ for any $r \in \mathbb{R}$. And we denote $G : (0, 1) \to \mathbb{R}$ which satisfies:

$$G(y) = F(\theta_1^{-1}(y)) \tag{57}$$

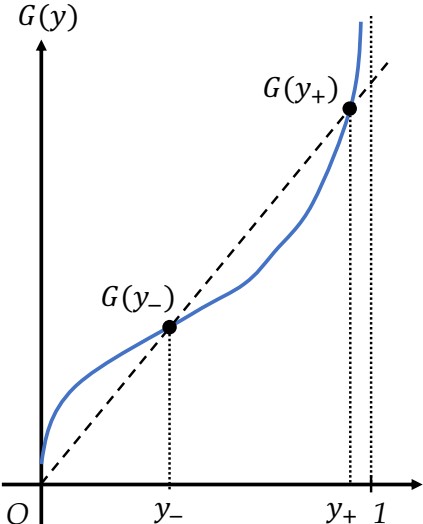

Figure 9: The plot of function $G(y)$.

and $y_+ := \theta_1^{-1}(r_+)$, $y_- := \theta_1^{-1}(r_-)$. Then if we can find some line $l_k : y = kx$ for some $k \in \mathbb{R}$ such that $l_k$ has at least two points of intersection $(y_i, ky_i), i = 1, 2$ with curve $G$ and $a_-/a_+ = y_1/y_2$ or $a_-/a_+ = y_2/y_1$, then we can always find some $\boldsymbol{v}$ and $\xi$ such that Eqn. 56 holds.

On the other hand, it's easy to find that (Fig. 9):

$$
\begin{aligned}
\frac{\mathrm{d}G(y)}{\mathrm{d}y}\Big|_{y=\theta_1(x)} &= \frac{\theta_1(x)}{p_n(x)} > 0 \\
\lim_{y \to 1} G(y) &= \lim_{r \to +\infty} F(r) = +\infty \\
\lim_{y \to 0} G(y) &= \lim_{r \to -\infty} F(r) = \lim_{r \to -\infty} r\theta_1(r)
\end{aligned}
$$

Note that since $G(y_+)/G(y_-) = y_+/y_-$, we have $G(y_+)/y_+ = G(y_-)/y_-$ and thus $(y_+, G(y_+))$ and $(y_-, G(y_-))$ are lying at the same straight line.

For finding the sufficient condition, we focus on the range $x \geq 0$ and $\theta_1(x) \geq \frac{1}{2}$. Then in order that line $l_k : y = kx$ for some $k \in \mathbb{R}$ has at least two points of intersection with curve $G$, we just need to let

$$
\frac{G(\tilde{\theta}_1(0))}{\tilde{\theta}_1(0)} \geq \frac{\mathrm{d}G(y)}{\mathrm{d}y}\Big|_{y=\tilde{\theta}_1(0)} \iff \tilde{\theta}_2(0) \cdot p_n(0) = p_n(0) \int_0^{+\infty} y' p_n(y')\mathrm{d}y' \geq \frac{1}{4} \tag{58}
$$

For convenience, let $S_{l_k} := \{(x, y)|y = kx\}$ and $S_G := \{(x, y)|y = G(x)\}$ to be the image of the needed functions. Denote $\pi_1 : \mathbb{R}^2 \to \mathbb{R} : \pi_1((x, y)) = x$ for any $x, y \in \mathbb{R}$, $\pi_1(S) = \{\pi_1(s)|\forall s \in S\}$. Therefore, if Eqn. 58 holds, then the following set $\mathcal{S}$ will not be empty.

$$
\mathcal{S} := \bigcup_{k \in \mathbb{R}} \{\frac{x_2}{x_1} \mid \forall x_1 \neq x_2 \in \pi_1(S_{l_k} \cap S_G)\} \tag{59}
$$

And Eqn. 5 has critical points if $a_+/a_- \in \mathcal{S}$. And it's easy to find that $\forall s \in \mathcal{S}$, $s \in (\frac{1}{2}, 1) \cup (1, 2)$. Similar results also hold for other homogeneous activations.

**Remarks.** It is often the case that $y_- < 1/2$ and $y_+ > 1/2$, since $G(y)$ when $y > 1/2$ is convex and there will be at most two intersection between a convex function and a straight line. This means that $r_+^* > 0$ and $r_-^* = \xi_* < 0$.

$\square$

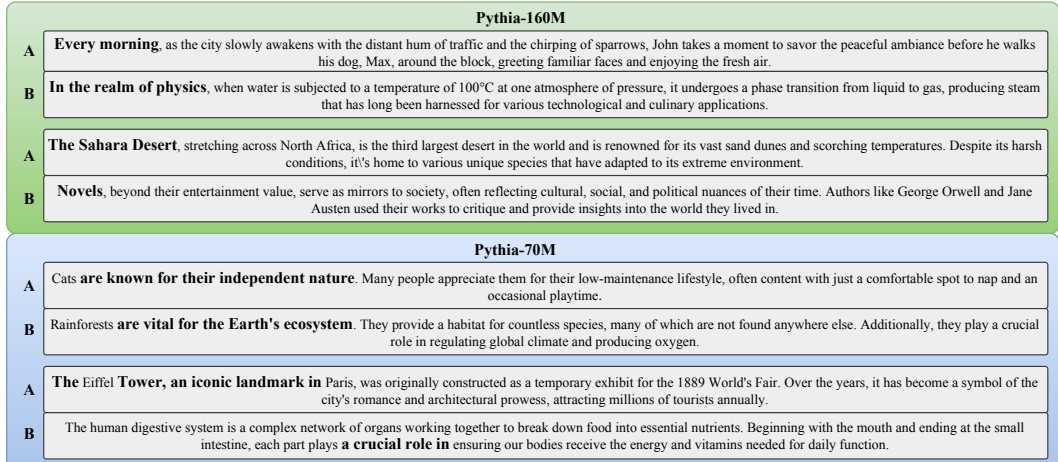

Figure 10: Examples of *pattern superposition*: the same neuron in MLP hidden layers can be activated by multiple irrelevant combinations of tokens (A and B in each group, e.g., the same neuron activated by both "Every morning" and "In the realm of physics"), in Pythia-70M and Pythia-160M models. Bold tokens are what the query token attends to.

## A.4  SEVERAL REMARKS

**The intuition behind $\xi$:** Note that while node $k$ in MLP layer does not have an explicit bias term, our analysis above demonstrates that there exists an "implicit bias" term $\xi_k(t)$ embedded in the weight vector $\boldsymbol{w}_k$:

$$\boldsymbol{w}(t) = \boldsymbol{w}(0) + U_C[\boldsymbol{v}(t) - \boldsymbol{v}(0)] + \boldsymbol{u}_m\xi(t) \tag{60}$$

This bias term allows encoding of the query embedding $\boldsymbol{u}_m$ into the weight, and the negative bias $\xi^* < 0$ ensures that given the query $q = m$, there needs to be a positive inner product between $\boldsymbol{v}_*$ (i.e., the "pattern template") and the input contextual tokens, in order to activate the node $k$.

**Pattern superposition.** Note that due to such mechanism, one single weight $\boldsymbol{w}$ may contain multiple query vectors (e.g., $\boldsymbol{u}_{m_1}$ and $\boldsymbol{u}_{m_2}$) and their associated pattern templates (e.g., $\boldsymbol{v}_{m_1}$ and $\boldsymbol{v}_{m_2}$), as long as they are orthogonal to each other. Specifically, if $\boldsymbol{w} = \boldsymbol{v}_{m_1} - \xi_{m_1}\boldsymbol{u}_{m_1} + \boldsymbol{v}_{m_2} - \xi_{m_2}\boldsymbol{u}_{m_2}$, then it can match both pattern 1 and pattern 2. We called this "pattern superposition", as demonstrated in Fig. 10.

**Lemma 4.** *If $\phi(x)$ is homogeneous, i.e., $\phi(x) = \phi'(x)x$, then there exist constant $c_-, c_+ \in \mathbb{R}$ depend on $\phi$ such that $\phi(x) = c_-\mathbf{1}[x < 0] + c_+\mathbf{1}[x > 0]$, and thus*

$$\frac{\mathrm{d}\theta_1}{\mathrm{d}r} = (c_- + c_+)p_n(r), \quad \frac{\mathrm{d}\theta_2}{\mathrm{d}r} = -(c_- + c_+)r \cdot p_n(r) \tag{61}$$

*Proof.* For any $x > 0$, we have

$$\phi'(x) = \lim_{\delta x \to 0+} \frac{\phi(x + \delta x) - \phi(x)}{\delta x} \tag{62}$$

$$= \lim_{\delta x \to 0+} \frac{\phi'(x + \delta x) - \phi'(x)}{\delta x} \cdot x + \lim_{\delta x \to 0} \phi'(x + \delta x) \tag{63}$$

$$= x \cdot \lim_{\delta x \to 0+} \frac{\phi'(x + \delta x) - \phi'(x)}{\delta x} + \phi'(x) \tag{64}$$

$$\tag{65}$$

So for any $x > 0$, $\phi'(x)$ must be constant, and similar results hold for $x < 0$. Then by direct calculation, we can get the results. □

A.4.1 WITH SELF-ATTENTION

**Lemma 5.** *Let* $g(y) := \frac{1-e^{-y^2}}{y}$. *Then* $\max_{y \geq 0} g(y) \leq \frac{1}{\sqrt{2}}$.

*Proof.* Any of its stationary point $y_*$ must satisfies $g'_y(y_*) = 0$, which gives:

$$e^{-y_*^2} = \frac{1}{2y_*^2 + 1} \tag{66}$$

Therefore, at any stationary points, we have:

$$g(y_*) = \frac{2y_*}{2y_*^2 + 1} = \frac{2}{2y_* + y_*^{-1}} \leq \frac{1}{\sqrt{2}} \tag{67}$$

since $g(0) = g(+\infty) = 0$, the conclusion follows. $\square$

**Lemma 6** (Bound of Gaussian integral). *Let* $G(y) := e^{-y^2/2} \int_0^y e^{x^2/2}\mathrm{d}x$, *then* $0 \leq G(y) \leq 1$ *for* $y \geq 0$.

*Proof.* $G(y) \geq 0$ is obvious. Note that

$$G(y) \quad := \quad e^{-y^2/2} \int_0^y e^{x^2/2}\mathrm{d}x \leq e^{-y^2/2} \int_0^y e^{xy/2}\mathrm{d}x = \frac{2}{y}\left(1 - e^{-y^2/2}\right) = \sqrt{2}g(y/\sqrt{2})$$

Applying Lemma 5 gives the conclusion. $\square$

**Theorem 4** (Convergence speed of salient vs. non-salient components). *Let* $\delta_j(t) := 1 - v_j(t)/\mu_j$ *be the convergence metric for component* $j$ *(*$\delta_j(t) = 0$ *means that the component* $j$ *converges). For nonlinear dynamics with attention (Eqn. 7), then*

$$\frac{\ln \delta_j(0)/\delta_j(t)}{\ln \delta_k(0)/\delta_k(t)} = \frac{e^{\mu_j^2/2}}{e^{\mu_k^2/2}}(1 + \Lambda_{jk}(t)) \tag{8}$$

*Here* $\Lambda_{jk}(t) = \lambda_{jk}(t) \cdot e^{\mu_k^2/2} \ln^{-1}(\delta_k(0)/\delta_k(t))$ *where* $|\lambda_{jk}(t)| \leq C_{jk}$ *and* $C_{jk}$ *only depends on* $\delta_j(0)$ *and* $\delta_k(0)$. *So when* $|\delta_k(t)| \ll |\delta_k(0)| \exp[-C_{jk} \exp(\mu_k^2)]$, *we have* $|\Lambda(t)| \ll 1$.

*Proof.* We first consider when $\boldsymbol{\mu} > 0$. We can write down the dynamics in a component wise manner, since all components share the same scalar constant:

$$\frac{\dot{v}_j}{\dot{v}_k} = \frac{(\mu_j - v_j)e^{v_j^2/2}}{(\mu_k - v_k)e^{v_k^2/2}} \tag{68}$$

which gives the following separable form:

$$\frac{\dot{v}_j e^{-v_j^2/2}}{\mu_j - v_j} = \frac{\dot{v}_k e^{-v_k^2/2}}{\mu_k - v_k} \tag{69}$$

Let

$$F(r, r_0, \mu) := \int_{r_0\mu}^{r\mu} \frac{e^{-v^2/2}}{\mu - v}\mathrm{d}v = \int_{r_0}^{r} \frac{e^{-\mu^2 x^2/2}}{1 - x}\mathrm{d}x \qquad (x = v/\mu) \tag{70}$$

Integrating both sides of Eqn. 69 from $t = 0$ to $t$, the dynamics must satisfy the following equation at time $t$:

$$F(r_j(t), r_j(0), \mu_j) = F(r_k(t), r_k(0), \mu_k) \tag{71}$$

where $r_j(t) := v_j(t)/\mu_j$. According to the dynamics, $r_j(t) \to 1$ and the question is how fast the convergence is. Depending on the initialization, $r_j(t) > 1$ or $r_j(t) < 1$.

Eqn. 71 implicitly gives the relationship between $r_j(t)$ and $r_k(t)$ (and thus $\delta_j(t)$ and $\delta_k(t)$). Now the question is how to bound $F(r, r_0, \mu)$, which does not have close-form solutions.

Note that we have:

$$\frac{\partial F}{\partial \mu} = -\mu \int_{r_0}^r \frac{x^2 e^{-\mu^2 x^2/2}}{1-x} \mathrm{d}x \tag{72}$$

$$= \mu \int_{r_0}^r \frac{1-x^2}{1-x} e^{-\mu^2 x^2/2} \mathrm{d}x - \mu \int_{r_0}^r \frac{e^{-\mu^2 x^2/2}}{1-x} \mathrm{d}x \tag{73}$$

$$= \mu \int_{r_0}^r (1+x) e^{-\mu^2 x^2/2} \mathrm{d}x - \mu F(r, r_0, \mu) \tag{74}$$

$$= \sqrt{\frac{\pi}{2}} \left[ \mathrm{erf}\left(\frac{r\mu}{\sqrt{2}}\right) - \mathrm{erf}\left(\frac{r_0\mu}{\sqrt{2}}\right) \right] + \frac{1}{\mu}(e^{-r_0^2\mu^2/2} - e^{-r^2\mu^2/2}) - \mu F(r, r_0, \mu) \tag{75}$$

Let

$$\zeta(r, r_0, \mu) := \sqrt{\frac{\pi}{2}} \left[ \mathrm{erf}\left(\frac{r\mu}{\sqrt{2}}\right) - \mathrm{erf}\left(\frac{r_0\mu}{\sqrt{2}}\right) \right] + \frac{1}{\mu}(e^{-r_0^2\mu^2/2} - e^{-r^2\mu^2/2}) \tag{76}$$

Applying Lemma 5 and notice that $\mu > 0$, we have

$$|\zeta(r, r_0, \mu)| \le \sqrt{2\pi} + \sqrt{2}(|r_0| + |r|)/\sqrt{2} \le \sqrt{2\pi} + \max(2|r_0|, |r_0| + 1) =: M(r_0) \tag{77}$$

which means that $|\zeta(r, r_0, \mu)|$ is uniformly bounded, regardless of $\mu$ and $r(t)$ (note that $r$ is bounded and will converge to 1 from the dynamics). Integrating both side and we have:

$$\frac{\partial}{\partial \mu}\left(e^{\mu^2/2} F(r, r_0, \mu)\right) = \zeta(r, r_0, \mu)e^{\mu^2/2} \tag{78}$$

$$e^{\mu^2/2} F(r, r_0, \mu) - F(r, r_0, 0) = \int_0^\mu \zeta(r, r_0, x)e^{x^2/2} \mathrm{d}x \tag{79}$$

$$F(r, r_0, \mu) = e^{-\mu^2/2} F(r, r_0, 0) + e^{-\mu^2/2} \int_0^\mu \zeta(r, r_0, x)e^{x^2/2} \mathrm{d}x \tag{80}$$

Note that $F(r, r_0, 0)$ has a close form:

$$F(r, r_0, 0) = \int_{r_0}^r \frac{1}{1-x} \mathrm{d}x = \ln \frac{1-r_0}{1-r} \tag{81}$$

has a close-form solution that works for both $r_0 < r < 1$ and $r_0 > r > 1$ (the situations that 1 is between $r_0$ and $r$ won't happen). Using mean-value theorem, we have:

$$F(r, r_0, \mu) = e^{-\mu^2/2} \ln \frac{1-r_0}{1-r} + \zeta(r, r_0, \bar{\mu})e^{-\mu^2/2} \int_0^\mu e^{x^2/2} \mathrm{d}x \tag{82}$$

Applying Lemma 6, we have the following bound for $F(r, \mu)$:

$$-M(r_0) \le F(r, \mu) - e^{-\mu^2/2} \ln \frac{1-r_0}{1-r} \le M(r_0) \tag{83}$$

When $r$ is close to 1 (near convergence), the term $e^{-\mu^2/2} \ln \frac{1-r_0}{1-r}$ (with fixed $\mu$ and fixed $r_0$) is huge compared to the constant $M(r_0)$, which is $\sqrt{2\pi} + 1.5 \approx 4.0066$ for e.g., $|r_0| = 1/2$, and thus $F(r, \mu) \to e^{-\mu^2/2} \ln \frac{1-r_0}{1-r}$.

To be more concrete, note that $\delta(t) = 1 - v(t)/\mu = 1 - r(t)$, we let

$$\rho(\delta(t), \mu) = F(1 - \delta(t), 1 - \delta(0), \mu) - e^{-\mu^2/2} \ln \frac{\delta(0)}{\delta(t)} \in (-M(r_0), M(r_0)) \tag{84}$$

And using Eqn. 71, we have:

$$F(1 - \delta_j(t), 1 - \delta_j(0), \mu_j) = F(1 - \delta_k(t), 1 - \delta_k(0), \mu_k) \tag{85}$$

Then

$$\lambda_{jk}(t) := \rho(\delta_k(t), \mu_k) - \rho(\delta_j(t), \mu_j) \tag{86}$$

$$= e^{-\mu_j^2/2} \ln \frac{\delta_j(0)}{\delta_j(t)} - e^{-\mu_k^2/2} \ln \frac{\delta_k(0)}{\delta_k(t)} \tag{87}$$

and $|\lambda_{jk}(t)| \le M(r_j(0)) + M(r_k(0))$. Then we arrive at the conclusion. $\qquad\square$

A.5 HIERARCHICAL LATENT TREE MODELS (SECTION 5)

We formally introduce the definition of `HBLT` here. Let $y_\alpha$ be a binary variable at layer $s$ (upper layer and $y_\beta$ be a binary variable at layer $s-1$ (lower layer). We use a 2x2 matrix $P_{\beta|\alpha}$ to represent their conditional probability:

$$P_{\beta|\alpha} := [\mathbb{P}[y_\beta|y_\alpha]] = \left[ \begin{array}{cc} \mathbb{P}[y_\beta = 0|y_\alpha = 0] & \mathbb{P}[y_\beta = 0|y_\alpha = 1] \\ \mathbb{P}[y_\beta = 1|y_\alpha = 0] & \mathbb{P}[y_\beta = 1|y_\alpha = 1] \end{array} \right] \tag{88}$$

**Definition 1.** *Define* $2 \times 2$ *matrix* $M(\rho) := \frac{1}{2} \left[ \begin{array}{cc} 1+\rho & 1-\rho \\ 1-\rho & 1+\rho \end{array} \right]$ *and 2-dimensional vector* $\boldsymbol{p}(\rho) = \frac{1}{2}[1+\rho, 1-\rho]^\top$ *for* $\rho \in [-1, 1]$.

**Lemma 7** (Property of $M(\rho)$)**.** *$M(\rho)$ has the following properties:*

- $M(\rho)$ *is a symmetric matrix.*

- $M(\rho)\boldsymbol{1}_2 = \boldsymbol{1}_2$.

- $M(\rho_1)M(\rho_2) = M(\rho_1\rho_2)$. *So matrix multiplication in* $\{M(\rho)\}_{\rho \in [-1,1]}$ *is communicative and isomorphic to scalar multiplication.*

- $M(\rho_1)\boldsymbol{p}(\rho_2) = \boldsymbol{p}(\rho_1\rho_2)$.

*Proof.* The first two are trivial properties. For the third one, notice that $M(\rho) = \frac{1}{2}(\boldsymbol{1}\boldsymbol{1}^T + \rho \boldsymbol{e}\boldsymbol{e}^\top)$, in which $\boldsymbol{e} := [1, -1]^\top$. Therefore, $\boldsymbol{e}^\top\boldsymbol{e} = 2$ and $\boldsymbol{1}^\top\boldsymbol{e} = 0$ and thus:

$$M(\rho_1)M(\rho_2) = \frac{1}{4}(\boldsymbol{1}\boldsymbol{1}^T + \rho_1\boldsymbol{e}\boldsymbol{e}^\top)(\boldsymbol{1}\boldsymbol{1}^T + \rho_2\boldsymbol{e}\boldsymbol{e}^\top) = \frac{1}{2}(\boldsymbol{1}\boldsymbol{1}^\top + \rho_1\rho_2\boldsymbol{e}\boldsymbol{e}^\top) = M(\rho_1\rho_2) \tag{89}$$

For the last one, note that $\boldsymbol{p}(\rho) = \frac{1}{2}(\boldsymbol{1} + \rho\boldsymbol{e})$ and the conclusion follows. $\square$

**Definition 2** (Definition of HBLT)**.** *In* `HBLT`$(\rho)$, $P_{\beta|\alpha} = M(\rho_{\beta|\alpha})$, *where* $\rho_{\beta|\alpha} \in [-1, 1]$ *is the uncertainty parameter. In particular, if* $\rho_{\beta|\alpha} = \rho$, *then we just write the entire* `HBLT` *model as* `HBLT`$(\rho)$.

**Lemma 8.** *For latent $y_\alpha$ and its descendent $y_\gamma$, we have:*

$$P_{\gamma|\alpha} = P_{\gamma|\beta_1}P_{\beta_1|\beta_2}\dots P_{\beta_k|\alpha} = M\left(\rho_{\gamma|\alpha}\right) \tag{90}$$

*where* $\rho_{\gamma|\alpha} := \rho_{\gamma|\beta_1}\rho_{\beta_1|\beta_2}\dots\rho_{\beta_k|\alpha}$ *and* $\alpha \succ \beta_1 \succ \beta_2 \succ \dots \succ \beta_k \succ \gamma$ *is the descendent chain from $y_\alpha$ to $y_\gamma$.*

*Proof.* Due to the tree structure of `HBLT`, we have:

$$\mathbb{P}[y_\gamma|y_\alpha] = \sum_{y_{\beta_1}, y_{\beta_2}, \dots, y_{\beta_k}} \mathbb{P}[y_\gamma|y_{\beta_1}]\mathbb{P}[y_{\beta_1}|y_{\beta_2}]\dots\mathbb{P}[y_{\beta_k}|y_\alpha] \tag{91}$$

which is precisely how the entries of $P_{\gamma|\beta_1}P_{\beta_1|\beta_2}\dots P_{\beta_k|\alpha}$ get computed. By leveraging the property of $M(\rho)$, we arrive at the conclusion. $\square$

**Theorem 5** (Token Co-occurrence in HBLT$(\rho)$)**.** *If token $l$ and $m$ have common latent ancestor (CLA) of depth $H$ (Fig. 5(c)), then* $\mathbb{P}[y_l = 1|y_m = 1] = \frac{1}{2}\left(\frac{1+\rho^{2H}-2\rho^{L-1}\rho_0}{1-\rho^{L-1}\rho_0}\right)$, *where $L$ is the total depth of the hierarchy and* $\rho_0 := \boldsymbol{p}_{\cdot|0}^\top\boldsymbol{p}_0$, *in which* $\boldsymbol{p}_0 = [\mathbb{P}[y_0 = k]] \in \mathbb{R}^D$ *and* $\boldsymbol{p}_{\cdot|0} := [\mathbb{P}[y_l = 0|y_0 = k]] \in \mathbb{R}^D$, *where $\{y_l\}$ are the immediate children of the root node $y_0$.*

*Proof.* Let the common latent ancestor (CLA) of $y_{\beta_1}$ and $y_{\beta_2}$ be $y_c$, then we have:

$$\mathbb{P}[y_{\beta_1}, y_{\beta_2}] = \sum_{y_c} \mathbb{P}[y_{\beta_1}|y_c]\mathbb{P}[y_{\beta_2}|y_c]\mathbb{P}[y_c] \tag{92}$$

Let $P_{\beta_1\beta_2} = [\mathbb{P}[y_{\beta_1}, y_{\beta_2}]]$, then we have:

$$P_{\beta_1\beta_2} = M(\rho_{\beta_1|c})D(c)M^\top(\rho_{\beta_2|c}) \tag{93}$$

where $D(c) := \mathrm{diag}(\mathbb{P}[y_c]) = \frac{1}{2}\begin{bmatrix} 1 + \rho_c & 0 \\ 0 & 1 - \rho_c \end{bmatrix}$ is a diagonal matrix, and $\rho_c := 2\mathbb{P}[y_c = 0] - 1$. Note that

$$\mathbf{1}^\top D(c)\mathbf{1} = \boldsymbol{e}^\top D(c)\boldsymbol{e} = 1, \qquad \mathbf{1}^\top D(c)\boldsymbol{e} = \boldsymbol{e}^\top D(c)\mathbf{1} = \rho_c \tag{94}$$

And $M(\rho) = \frac{1}{2}(\mathbf{1}\mathbf{1}^T + \rho \boldsymbol{e}\boldsymbol{e}^\top)$, therefore we have:

$$
\begin{aligned}
P_{\beta_1 \beta_2} &= M(\rho_{\beta_1|c})D(c)M^\top(\rho_{\beta_2|c}) \tag{95} \\
&= \frac{1}{4}(\mathbf{1}\mathbf{1}^T + \rho_{\beta_1|c}\boldsymbol{e}\boldsymbol{e}^\top)D(c)(\mathbf{1}\mathbf{1}^T + \rho_{\beta_2|c}\boldsymbol{e}\boldsymbol{e}^\top) \tag{96} \\
&= \frac{1}{4}\left(\mathbf{1}\mathbf{1}^T + \rho_{\beta_1|c}\rho_{\beta_2|c}\boldsymbol{e}\boldsymbol{e}^\top + \rho_{\beta_1|c}\rho_c\boldsymbol{e}\mathbf{1}^\top + \rho_{\beta_2|c}\rho_c\mathbf{1}\boldsymbol{e}^\top\right) \tag{97}
\end{aligned}
$$

Now we compute $\rho_c$. Note that

$$\mathbb{P}[y_c] = \sum_{y_0}\mathbb{P}[y_c|y_0]\mathbb{P}[y_0] \tag{98}$$

Let $\boldsymbol{p}_c := [\mathbb{P}[y_c]]$ be a 2-dimensional vector. Then we have $\boldsymbol{p}_c = P_{y_c|y_0}\boldsymbol{p}_0 = \boldsymbol{p}(\rho_{c|0}\rho_0)$, where $\boldsymbol{p}_0$ is the probability distribution of class label $y_0$, which can be categorical of size $C$:

$$
\begin{aligned}
\boldsymbol{p}_c &= P_{y_c|y_0}\boldsymbol{p}_0 = \sum_{y_1} P_{y_c|y_1}P_{y_1|y_0}\boldsymbol{p}_0 \tag{99} \\
&= M(\rho_{c|1})\frac{1}{2}\begin{bmatrix} 1 + p_{1|0} & 1 + p_{2|0} & \cdots & 1 + p_{C|0} \\ 1 - p_{1|0} & 1 - p_{2|0} & \cdots & 1 - p_{C|0} \end{bmatrix}\boldsymbol{p}_0 \tag{100} \\
&= M(\rho_{c|1})\frac{1}{2}\begin{bmatrix} 1 + \boldsymbol{p}_{\cdot|0}^\top\boldsymbol{p}_0 \\ 1 - \boldsymbol{p}_{\cdot|0}^\top\boldsymbol{p}_0 \end{bmatrix} \tag{101} \\
&= M(\rho_{c|1}\boldsymbol{p}_{\cdot|0}^\top\boldsymbol{p}_0) \tag{102}
\end{aligned}
$$

in which $y_1$ is the last binary variable right below the root node class label $y_0$.

Therefore, $\rho_c = \rho_{c|1}\rho_0$, where $\rho_0 := \boldsymbol{p}_{\cdot|0}^\top\boldsymbol{p}_0$ is the uncertainty parameter of the root node $y_0$.

If all $\rho_{\beta|\alpha} = \rho$ for immediate parent $y_\alpha$ and child $y_\beta$, $y_{\beta_1}$ is for token $l$ and $y_{\beta_2}$ is for token $m$, then $\rho_{\beta_1|c} = \rho_{\beta_2|c} = \rho^H$, and $\rho_{c|1} = \rho^{L-1-H}$ and thus we have:

$$
\begin{aligned}
\mathbb{P}[y_l = 1|y_m = 1] &= \frac{\mathbb{P}[y_l = 1, y_m = 1]}{\mathbb{P}[y_m = 1]} = \frac{1}{2}\left(\frac{1 + \rho^{2H} - 2\rho^H\rho_c}{1 - \rho^H\rho_c}\right) \tag{103} \\
&= \frac{1}{2}\left(\frac{1 + \rho^{2H} - 2\rho^{L-1}\rho_0}{1 - \rho^{L-1}\rho_0}\right) \tag{104}
\end{aligned}
$$

and the conclusion follows. $\qquad\square$

## B  MORE EXPERIMENT RESULTS

### B.1  ORTHOGONALITY OF EMBEDDING VECTORS

We verify the orthogonality assumption mentioned in our problem setting (Sec. 2). The orthogonality is measured by absolute cosine similarity $\mathrm{cossim}(\boldsymbol{x}_1, \boldsymbol{x}_2) \in [0, 1]$ of two vectors $\boldsymbol{x}_1$ and $\boldsymbol{x}_2$:

$$\mathrm{cossim}(\boldsymbol{x}_1, \boldsymbol{x}_2) := \frac{|\boldsymbol{x}_1^\top\boldsymbol{x}_2|}{\|\boldsymbol{x}_1\|\|\boldsymbol{x}_2\|} \tag{105}$$

Here the two vectors $\boldsymbol{x}_1$ and $\boldsymbol{x}_2$ are column vectors of the out-projection (or upper) matrix of MLPs at different layers, each corresponding to one hidden neuron. For a MLP layer with model dimension $d$ and hidden dimension $4d$, there will be $4d$ such column vectors. We measure the average cosine similarity across all $2d(4d - 1)$ pairs and report in the figure.

While $4d$ $d$-dimensional vectors have to be linearly dependent, they are indeed almost orthogonal (i.e., $\text{cossim}(\boldsymbol{x}_1, \boldsymbol{x}_2) \ll 1$) throughout the training process, as shown below. In Fig. 11, we show cosine similiarity over the entire training process of Pythia models of different sizes. Fig. 12 further checks the training curve at early training stages, since Pythia checkpoints are more densely sampled around early training stages, i.e., "steps 0 (initialization), 1, 2, 4, 8, 16, 32, 64, 128, 256, 512, 1000, and then every 1,000 subsequent steps" (Biderman et al., 2023). Finally, for models whose intermediate checkpoints are not available, we show the cosine similarity in the publicly released pre-trained models (Fig. 13).

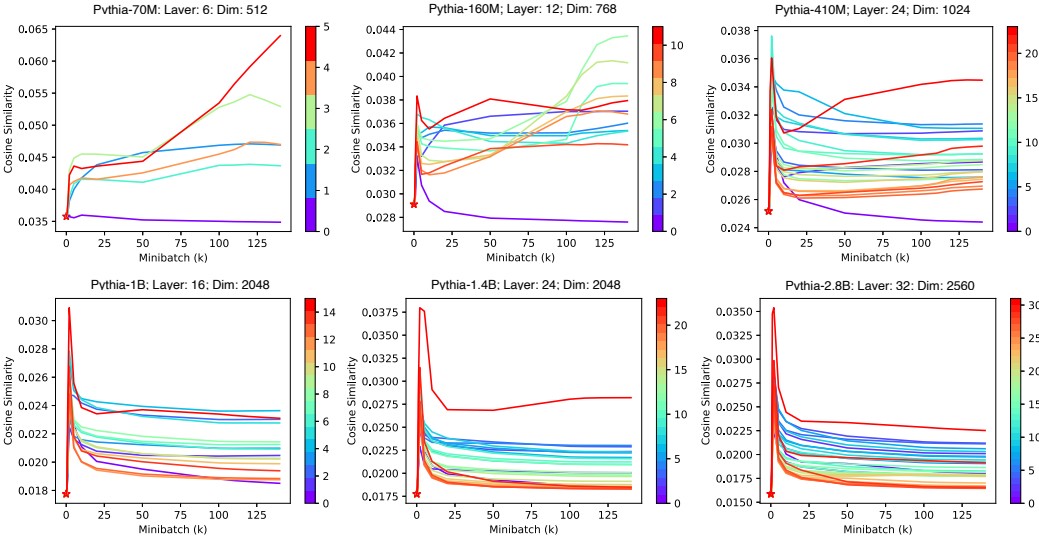

Figure 11: Orthogonality of embeddings of MLP in LLMs during the whole training process.

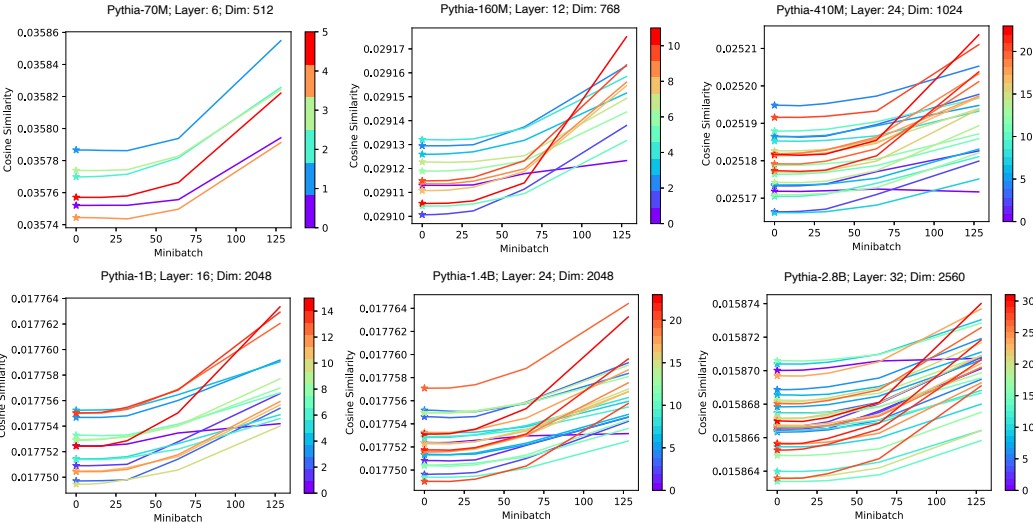

Figure 12: Orthogonality of embeddings of MLP in LLMs during the early training stage.

## B.2 ATTENTION ENTROPY FOR ENCODER-DECODER MODELS

We also measure how attention entropy, as well as stable rank of the in-projection (or lower) matrix in MLP, changes over time for encoder-decoder models like BERT, as shown in Fig. 14. The behavior is very similar to the decoder-only case (Fig. 7), further verifying our theoretical findings.

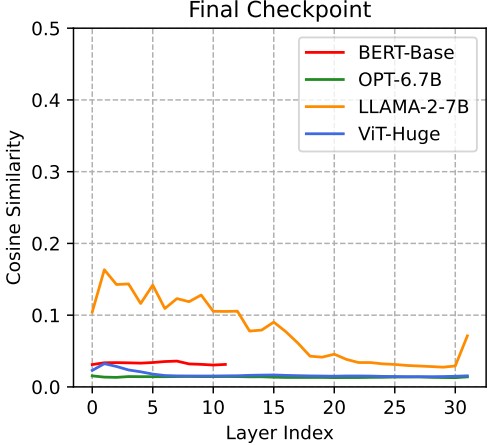

Figure 13: Orthogonality measures in model architectures (BERT-Base, OPT-6.7B, LLaMA-2-7B, ViT-Huge), with only final checkpoint available.

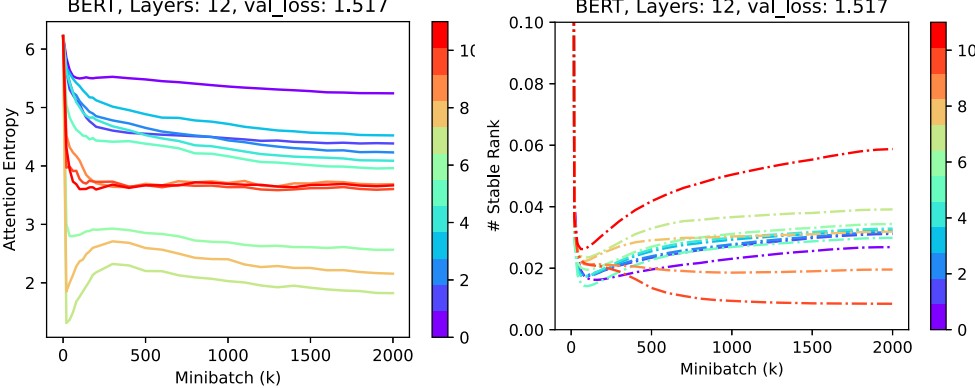

Figure 14: (Left) Attention entropy of BERT; (Right) Stable rank in BERT.

