# OpenReview forum: "JoMA: Demystifying Multilayer Transformers via Joint Dynamics of MLP and Attention"
_ICLR.cc/2024/Conference — ICLR 2024 poster_

### Official Review · Reviewer_5uGr · 2023-10-23

**Soundness:** 3 good
**Presentation:** 3 good
**Contribution:** 3 good
**Rating:** 5
**Confidence:** 2

**Summary:**

The paper introduces Joint MLP/Attention (JoMA) dynamics, a mathematical framework designed to understand the training process of multilayer Transformer architectures. This is accomplished by integrating out the self-attention layer in Transformers, resulting in a modified dynamics that focuses solely on the MLP layers. JoMA addresses limitations present in previous analyses by removing unrealistic assumptions, such as the lack of residual connections.

The authors predict that attention in JoMA initially becomes sparse to learn salient tokens and then transitions to dense attention to capture less salient tokens when nonlinear activations are incorporated. In the case of linear activations, the predictions align with existing literature. Additionally, JoMA is leveraged to provide qualitative explanations of how tokens combine to form hierarchies in multilayer Transformers, specifically in scenarios where input tokens are generated by a latent hierarchical generative model.

To validate their theoretical findings, the authors conduct experiments using real-world datasets (Wikitext2/Wikitext103) and various pre-trained models (OPT, Pythia). The experimental results support their claims and contribute to the credibility of their research.

**Strengths:**

+ The paper introduces the JoMA dynamics, which is a new mathematical framework specifically designed to understand the training procedure of multilayer Transformer architectures.

+ The authors highlight that previous analyses of similar models often make unrealistic assumptions, such as the lack of residual connections. In contrast, the JoMA framework removes these assumptions, making it more accurate and realistic in capturing the training dynamics of multilayer Transformers.

**Weaknesses:**

- Although the paper mentions experimental validation using real-world datasets and pre-trained models, it does not provide detailed quantitative evaluation metrics or comparisons against existing approaches. Including quantitative analysis would enhance the robustness and comprehensiveness of the findings.

- The paper does not delve into specific implementation details of the JoMA framework, such as hyperparameter choices or training procedures. Providing more information on these aspects would enable researchers to replicate the experiments and further validate the proposed approach.

**Questions:**

1. Why is Figure 1(b) labeled as the "problem setting"?

---

> ### Author Response · Authors · 2023-11-17
> **Rebuttal**
>
> We thank the reviewer for the insightful and encouraging comments!
>
> Note that the reviewer may misunderstand our main contribution. We do not propose a novel approach to push SoTA, but instead we analyze existing Transformer architectures via a novel angle, by finding the joint dynamics of the self-attention and the MLP layer and using these new mathematical finding to analyze the training dynamics of multi-layer Transformer architectures. Therefore, there is no comparison against existing approaches.
>
> We indeed provide detailed quantitative analysis to show our theoretical findings are consistent with the real world scenarios. This includes the following:
> + We plot the change of entropy of attention patterns over time at different layers (Fig.6-8), as well as changes of stable rank in MLP layers (Fig. 7), verifying our theoretical findings (Sec. 4 and Theorem 4).
> + We show that the hidden neurons in the Transformer MLP layer indeed learn the latent concepts in the hierarchy (Tbl. 1).
> + We have multiple experiments that verify our intermediate theoretical findings. For example, Fig. 2 shows that Theorem 1 largely holds for softmax attention, even if the assumption for the theorem to hold for softmax may not be that realistic. Fig. 3 verifies Theorem 2 (growth in weights in the case of linear activations).
>
> Since JoMA is not a novel approach but a way to analyze existing Transformer architectures with more realistic assumptions, we do not have specific implementation details of the framework like hyper-parameter choices. We follow the standard practice in training and mention our choices of hyper-parameters in Sec. 6.
>
> Fig. 1(b) is labeled as "problem setting" since this is the specific, mathematically well-defined Transformer architecture we analyze in JoMA.

---

### Official Review · Reviewer_cMvf · 2023-10-31

**Soundness:** 3 good
**Presentation:** 3 good
**Contribution:** 3 good
**Rating:** 6
**Confidence:** 3

**Summary:**

This paper proposes a novel mathematical framework called JoMA to understand the training procedure of multilayer Transformer architectures. By integrating out the self-attention layer in Transformers, the authors focus on the dynamics of MLP layers. The framework removes unrealistic assumptions and provides insights into the behavior of attention mechanisms. The authors leverage JoMA to explain how tokens form hierarchies in multilayer Transformers using a latent hierarchical generative model. Experimental results on real-world datasets and pre-trained models validate the theoretical findings.

**Strengths:**

1. The proposed JoMA framework offers a new perspective on understanding the training dynamics of multilayer Transformers. By integrating out the self-attention layer, the authors provide valuable insights into the behavior of MLP layers.

2. This paper presents theoretical findings that explain how tokens are combined to form hierarchies in multilayer Transformers. These findings are supported by experiments on real-world datasets and pre-trained models, enhancing the credibility of the results.

3. This paper is well-written and structured, making it easy to follow the proposed framework and understand the theoretical explanations. The inclusion of references to related works in the field strengthens the paper's contribution.

**Weaknesses:**

1. Limited practical implications. This paper focuses on theoretical analysis and understanding the training dynamics of multilayer Transformers. It would be valuable to discuss potential practical applications or implications of the proposed framework in real-world scenarios.

2 Several lines of work around the learning preference of DNNs need to discussed [cite1-3].

[cite1] Devansh Arpit, Stanisław Jastrz˛ebski, Nicolas Ballas, David Krueger, Emmanuel Bengio,
Maxinder S Kanwal, Tegan Maharaj, Asja Fischer, Aaron Courville, Yoshua Bengio, et al. A
closer look at memorization in deep networks. In International conference on machine learning,
pages 233–242. PMLR, 2017
[cite2] Karttikeya Mangalam and Vinay Uday Prabhu. Do deep neural networks learn shallow learnable
examples first? 2019.
[cite3] Huh M, Mobahi H, Zhang R, et al. The low-rank simplicity bias in deep networks[J]. arXiv:2103.10427, 2021.

3  I am not sure how the insights from this paper could help us train/design better DNNs.

**Questions:**

Please see Weaknesses.

---

> ### Author Response · Authors · 2023-11-17
> **Rebuttal**
>
> We thank he reviewer for the insightful feedbacks!
>
> # Comparison with existing works regarding learning preference of DNN
>
> We thank the reviewer for providing these related work and will include the following comparison in our next revision.
>
> Both [cite1] and [cite2] focus on empirical study of the training dynamics of DNNs (not transformers) on real data vs random data, following the seminar paper “Understanding deep learning requires rethinking generalization” in ICLR’17.
>
> [cite1] shows that DNN will learn simple data points/patterns first, when training with real data, while for random data (either random input or random label), each data sample is equally difficult. It also studies the effects of different regularization on training curves on real/random data. [cite2] further shows that the concept of “simple samples/patterns” generalizes across multiple architectures and learning methods (e.g., SVM versus DNN).
>
> Note that both JoMA and [cite1][cite2] conclude that simple patterns are learned first, followed by learning of more complicated patterns. Both [cite1] and [cite2] treat DNNs as a black box and provide thorough empirical studies. They also do not analyze Transformers. In contrast, focusing on Transformer architecture that shows impressive performance across multiple domains, JoMA opens the architecture blackbox and gives more quantitative definition of patterns in terms of co-occurrence, and explains how the specific architecture of Transformer learns these patterns by implicitly learning a latent hierarchy.
>
> Furthermore, we could also relate our analysis to [cite1][cite2], by re-defining the somehow vague term “simple/difficult patterns” with more rigorous quantity: “simple patterns” can be co-occurred tokens in the lowest layer of hierarchy, while “difficult patterns” may come across multiple layers of hierarchy. In this case, our JoMA framework naturally explains why simple patterns are learned first, followed by difficult patterns. It also explains why the “easiness of pattern” translates from SVM to more complicated architectures: since co-occurred tokens in the lowest hierarchy layer can be easily learned by linear models.
>
> [cite3] focuses on empirical study of low (effective) rank bias of DNNs after training. It shows that low-rank bias happens in different optimizers and is insensitive to initializations. Compared to [cite1] and [cite2], [cite3] focuses less on the training dynamics but more on the property of the final trained model. In comparison, from the training dynamics point of view, JoMA not only provides a theoretical justification why such phenomena (i.e., emergence of low-rank structure) could happen during training, but also characterizes the change of ranks in a more refined way (i.e., high rank -> low rank -> high rank again) and discusses the underlying reason why the learning is done in this way (i.e., to learn salient feature first at lower layer, leaving non-salient features at the top layer of the hierarchy).
>
> # How the insights from JoMA help with practical applications for real-world scenarios
> JoMA focuses on specific architectures and can provide meaningful suggestions on how the architectures can be changed. Here are a few examples:
> + Theorem 1 suggests that certain designs of soft-attention activation function lead to the invariance and future attention designs may be inspired by that.
> + The attention entropy (and stable rank) re-bouncing curves suggest that low-rank structure may not be present over the entire training procedure. This suggests that we may want to use high-rank weight matrices at the beginning and the end of the training era.
> + Fig. 8 shows that the re-bouncing of attention entropy over time is highly correlated with low validation error of the model after training. This suggests that we could check whether a run is healthy by checking the attention curve.
> + From Sec. 5, we know that the learning of non-salient co-occurrence is slow, because the model is unsure whether they should be learned under the current hierarchy, or in the higher hierarchy. Knowing this, if we have prior knowledge on the structure of the feature hierarchy, we may develop novel algorithms to accelerate the training.
>
> Reviewer **h5HX** also appreciates our work and thinks our understanding of hierarchical learning via self-attention can guide the design of follow-up transformer-based networks.
>
> Note that overall, the main focus of JoMA is to understand how multi-layer Transformer works and provide theoretical insights for the community, rather than improving practical applications for real-world scenarios.

---

> > ### Comment · Reviewer_cMvf · 2023-11-22
> >
> > Thanks for the response. The revision can improve this paper. I will keep my score.

---

### Official Review · Reviewer_h5HX · 2023-10-31

**Soundness:** 3 good
**Presentation:** 4 excellent
**Contribution:** 3 good
**Rating:** 6
**Confidence:** 3

**Summary:**

This paper named JoMA, explores to explain the training procedure of multi-layer transformers. JoMA delivers a conclusion that transformers learn salient tokens at low layers while learning less salient tokens at high layers when the nonlinear activations involve training. To build a mathematical framework to show the learning mechanism of multi-layer transformer, JoMA proposes joint dynamics of self-attention and MLP.

Compared with former works that focus on shallow transformer networks, linear activation or local gradient steps, JoMA builds a unified mathematical framework to characterize the learning mechanism of multi-layer transformers. Analyzing deep transformer networks is more challenging.

**Strengths:**

JoMA is able to analyze deep and sophisticated transformer networks.

It's interesting to use dynamics to unify the MLP, self-attention, and non-linear activation.

The derived conclusion of how self-attention learns hierarchical data distribution is meaningful and can guide the design of follow-up transformer-based network.

**Weaknesses:**

It would be better to discuss the limitation of JoMA.

It's not clear why chose OPT and Pythis for verification. Can the findings in this work coincide with BERT-style models?

**Questions:**

-

---

> ### Author Response · Authors · 2023-11-17
> **Rebuttal**
>
> We thank the reviewer for the insightful feedbacks!
>
> # Can the findings in this work coincide with BERT-style models?
> We choose OPT and Pythia because they provide public intermediate checkpoints so that we can check the training dynamics.
>
> Theoretically our analysis can be extended to BERT like models (or encoder-decoder models). Empirically, We have done experiments on BERT and the conclusion is similar. The attention entropy, as well as the stable rank, has the bounce-back behavior over time (see Appendix B.2 in our revised paper).
>
> # Limitation of JoMA
> JoMA has the following limitations:
> + We assume the back-propagated gradients are stationary (or slowly changing over time).
> + We assume the embedding vectors are orthogonal and fixed during training.
> + We provide qualitative analysis for hierarchical latent generative models, but not quantitative.
> + We also do not analyze how the model size affects the learning procedure.
>
> Note that these limitations are necessary assumptions to make the main message of this paper concise. Some substantial amount of work is needed to remove these assumptions, which is beyond the scope of this work. We will address them in our future work.

---

> > ### Comment · Reviewer_h5HX · 2023-11-20
> > **Response**
> >
> > Thanks for the rebuttal.
> >
> > The authors have addressed my concerns and I will maintain my positive score.

---

### Official Review · Reviewer_P4nM · 2023-10-31

**Soundness:** 3 good
**Presentation:** 3 good
**Contribution:** 3 good
**Rating:** 6
**Confidence:** 2

**Summary:**

The paper proposes a mathematical framework called JoMA to understand the training dynamics of multilayer Transformers. JoMA integrates out the self-attention layer and produces a modified dynamics of MLP layers only. JoMA reveals the training dynamics under linear or non-linear attention, as well as how the attention sparsity change over time, and how they depend on the input distribution.. The paper also verifies its theoretical findings with experiments on real-world data.

**Strengths:**

By integrating out the self-attention layer and analyzing only the modified dynamics of MLP layers, JoMA provides a fresh perspective on how these models learn and adapt during training. The paper demonstrates rigorous theoretical analysis on training dynamics, especially on how attention evolves over time. An interesting observation from the paper is that MLP and attention shares the same dynamics, which leads to several intriguing conclusions, for example, the rank of lower mlp layers first goes down and then bounces up. The mathematical derivations are presented clearly, making it accessible to researchers with different backgrounds.

**Weaknesses:**

1. One assumption in the paper is that the embedding vectors of all tokens are orthogonal to each other. Does this need to hold for token embedding in every layer? Have the authors verified this assumption is valid in real cases? How much does this assumption affect the following conclusions?

**Questions:**

1. This paper analyze the training dynamics of multi-layer transformer and reveals how the attention is learned through time (first learning attention on tokens with most co-occurrence, and then expanding attention on other tokens). One question is, how is this dynamics dependent on model size? As we see from empirical results in the literature, larger model tends to learn attention that involves more complicated reasoning instead of simple co-occurrence. Can this factor be reflected by the analysis in the paper? How does the training dynamics (and entropy of attention) change when model size changes?

2. Is the proposed JoMA also applicable to vision transformers? Since the input to vision transformer is different from text transformer in that the tokens are continuous instead of discrete and several assumptions in the paper may not hold. For example, the paper assumes tokens are orthogonal to each other, but for vision input lots of input tokens are very similar.

---

> ### Author Response · Authors · 2023-11-17
> **Rebuttal**
>
> We thank the reviewer for the insightful feedback!
>
> # Orthogonality assumption of embedding vectors
> In order to reach concise conclusions, we made this assumption. The assumption needs to be true for all embedding layers (i.e., the upper out-projection matrix of MLP at every layer), otherwise there will be many additional terms in our analysis, and we won’t be able to give a clear overall picture of the training.
>
> In practice, we show that the embedding vectors are almost orthogonal to each other in many models. For Pythia models of different sizes (from 70M to 6.9B), the averaged absolute cosine similarity over all embedding vector pairs is bounded by 0.1 throughout the training, much smaller than the maximal value 1. For other pre-trained models like BERT, OPT-6.7B and ViT-Huge, it is also small (<0.05), for LLaMA-2-7B, it is <0.17. Please check Appendix B.1 for these additional experiments in our revised submission.
>
> Therefore, our analysis largely holds, but may contain many small terms. A detailed analysis with almost orthogonal vectors will be left for future work.
>
> Note that for VIT, we measure the embedding layer (i.e., the upper out-projection matrix of MLP at every layer), but not the first input layer, which may have very strong correlations between nearby embeddings.
>
> # How larger models tend to learn attention that involves more complicated reasoning instead of simple co-occurrence
> For this, we break the questions into two parts:
> + How do Transformers learn more complicated reasoning than simple co-occurrence?
> + How do large models help?
>
> [**Learning more complicated reasoning**] In this work, we indeed focus on feature composition based on co-occurrence, which is a simple form (maybe the simplest form) of feature hierarchy. For reasoning, an intuition is that the model still captures co-occurring patterns. The difference is that thanks to embedding representation, these patterns are combinations of certain learnable “attributes” of input data, rather than input data themselves.
>
> For example, from the dataset “a x 3 = a a a”, “b x 3 = b b b”, the model will predict “c c c” from “c x 3 =”. Why can this be learned? From our point of view, there could be two critical components that contribute to this process:
> + The embedding of “a”, $u_a$, can contain multiple aspects of the letter “a”, e.g., $u_a = u_{[alphabet]} + u_{[first]}$, which means that a is “the first letter in the alphabet”. In this case, The pattern “[alphabet] x 3” is a common co-occurred pattern that can be extracted from “a x 3” and “b x 3”.
> + As JoMA suggests, some neuron in the MLP hidden layer may represent the pattern p = “[alphabet] x 3”, and use an embedding to represent it for the next layer. Due to the residual connection, the resulting representation of “a x 3” can be $u_{p} + u_{[alphabet]} + u_{[first]}$.
> + The co-occurred pattern $u_p + u_{[first]}$ will have a large inner product with the embedding “a” in the decoder layer, which also has a $u_{[first]}$ component. As a result, the letter “a” is decoded with high probability. Note that this procedure can generalize to any letter (e.g., c or d), as long as their embedding takes the form of $u_{[alphabet]} + …$.
>
> In summary, for reasoning tasks:
> + Learning co-occurred patterns of the attributes that summarize different cases, by leveraging the power of embeddings.
> + The matched pattern is also represented as embedding, superimposed to the original embeddings, thanks to residual connections.
> Overall, this gives very efficient representations and avoids enumerating all possible instantiation of patterns. Note that this is beyond JoMA's scope and we will leave a more mathematically rigorous study as the future work.
>
> [**How do large models help?**] Large model helps in the following two ways:
> + Larger models typically have more hidden neurons in the MLP layers. According to lottery ticket hypothesis[1], it is more likely that there exists at least one hidden neuron that happens to initialize with a weight vector and an embedding that is useful for modeling a compositional concept. Such a good initialization will facilitate training.
> + Larger models typically have larger feature dimension d, as suggested in Sec. 7 (Johnson–Lindenstrauss lemma), substantially more embedding vectors can be placed into a space with larger d, while maintaining almost orthogonality.
>
> Note that JoMA is just a starting point that captures the simplest mechanism of feature compositions. To make our analysis more straightforward and easier to follow, we also assume all embeddings are fixed/orthogonal during training. A thorough and more rigorous theoretical study of how Transformer performs reasoning, and how the embedding vectors are learned during training, is left for future work.

---

> > ### Comment · Reviewer_P4nM · 2023-11-21
> > **Thank you for the response**
> >
> > I appreciate the detailed response from the authors. I will remain my positive score.

---

### Author Response · Authors · 2023-11-17
**Rebuttal**

We sincerely thank the reviewers for their insightful feedback.

We are glad to hear that reviewers all appreciate that the paper focuses on multi-layer Transformers which is more challenging (**h5HX**) than 1-layer, and addresses limiting assumptions of previous theoretical works that analyze Transformers (**5uGr**).

In term of main contributions,
+ JoMA shows the existence of joint dynamics of MLP and attention layers (**P4nM**, **h5HX**, **cMvf**, **5uGr**)
+ JoMA shows how the attention evolves over time, in particular, its entropy goes down and then bumps up (**P4nM**).
+ JoMA shows that the salient features learn first, followed by non-salient ones (**h5HX**, **5uGr**), and qualitatively characterizes the feature compositionality process under hierarchical latent generative models (**h5HX**, **cMvf**, **5uGr**).

In terms of clarity, our work conveys clear high-level messages (**cMvf**) and mathematical derivations (**P4nM**). We also provide experiments on real-world datasets and pre-trained models to verify the theoretical findings (**cMvf**, **5uGr**).

Here we list a few general concerns and answer here.

# Assumption and limitations

JoMA has the following limitations
+ We assume the back-propagated gradients are stationary (or slowly changing over time).
+ We assume the embedding vectors are orthogonal and fixed during training.
+ We only provide qualitative analysis for hierarchical latent generative models, but not quantitative.
+ We also do not analyze how the model size affects the learning procedure.

Note that these limitations are necessary assumptions to make the main message of this paper concise. Some substantial amount of work is needed to remove these assumptions, which is beyond the scope of this work. We will address them in our future work.

[**Orthogonality assumption of embedding vectors**] In order to reach concise conclusions, we made this assumption. The assumption needs to be true for all embedding layers (i.e., the upper out-projection matrix of MLP at every layer), otherwise there will be many additional terms in our analysis, and we won’t be able to give a clear overall picture of the training.

In practice, we show that the embedding vectors are almost orthogonal to each other in many models. For Pythia models of different sizes (from 70M to 6.9B), the averaged absolute cosine similarity over all embedding vector pairs is bounded by 0.1 throughout the training, much smaller than the maximal value 1. For other pre-trained models like BERT, OPT-6.7B and ViT-Huge, it is also small (<0.05), for LLaMA-2-7B, it is <0.17. Please check Appendix B.1 for these additional experiments in our revised submission.

Therefore, our analysis largely holds, but may contain many small terms. A detailed analysis with almost orthogonal vectors will be left for future work.

Note that for VIT, we measure the embedding layer (i.e., the upper out-projection matrix of MLP at every layer), but not the first input layer, which may have very strong correlations between nearby embeddings.

[**How do large models help?**] Large model helps in the following two ways:
+ Larger models typically have more hidden neurons in the MLP layers. According to lottery ticket hypothesis[1], it is more likely that there exists at least one hidden neuron that happens to initialize with a weight vector and an embedding that is useful for modeling a compositional concept. Such a good initialization will facilitate training.
+ Larger models typically have larger feature dimension d, as suggested in Sec. 7 (JL lemma), substantially more embedding vectors can be placed into a space with larger d, while maintaining almost orthogonality.

Note that these are intuitions and we will leave thorough and rigorous study as the future work.

# Application of the conclusion to encoder-decoder types of models
We have done experiments on BERT and the conclusion is similar. The attention entropy, as well as the stable rank, has the bounce-back behavior over time (see Appendix B.2 in our revised paper).

---

> ### Author Response · Authors · 2023-11-17
> **Rebuttal (cont.)**
>
> # Application to real world scenarios
> JoMA focuses on specific architectures and can provide meaningful suggestions on how the architectures can be changed. Here are a few examples:
> + Theorem 1 suggests that certain designs of soft-attention activation function lead to the invariance and future attention designs may be inspired by that.
> + The attention entropy (and stable rank) re-bouncing curves suggest that low-rank structure may not be present over the entire training procedure. This suggests that we may want to use high-rank weight matrices at the beginning and the end of the training era.
> + Fig. 8 shows that the re-bouncing of attention entropy over time is highly correlated with low validation error of the model after training. This suggests that we could check whether a run is healthy by checking the attention curve.
> + From Sec. 5, we know that the learning of non-salient co-occurrence is slow, because the model is unsure whether they should be learned under the current hierarchy, or in the higher hierarchy. Knowing this, if we have prior knowledge on the structure of the feature hierarchy, we may develop novel algorithms to accelerate the training.
>
> Reviewer **h5HX** also appreciates our work and thinks our understanding of hierarchical learning via self-attention can guide the design of follow-up transformer-based networks.
>
> Note that overall, the main focus of JoMA is to understand how multi-layer Transformer works and provide theoretical insights for the community, rather than improving practical applications for real-world scenarios.

---

### Meta-Review · Area_Chair_xxXV · 2023-12-07

**Metareview:**

This paper investigates a theoretical framework to understand the training dynamics of Transformers. To this end, the authors propose to integrate the lower MLP and self-attention into a single joint MLP. This allows them to analyze their joint dynamics during training under various attention kernels and multi-layer settings. The analysis leads to several interesting findings such as how the sparsity of attention changes over time and how the tokens are combined to form hierarchies in multi-layer Transformers.

The paper received borderline scores of three borderline accepts and one borderline reject. The concerns raised by the reviewers were about the validity of assumptions, such as orthonormal embedding vectors, and the applicability of the JoMA to bidirectional Transformers such as BeRT or ViT. The authors adequately addressed most concerns by the reviewers in rebuttal.

After reading the paper, reviews, and rebuttal, AC believes that the paper indeed provides meaningful insights on training dynamics on Transformers, and provides convincing evidence with existing Transformer-based models. The authors should incorporate additional discussions presented in the rebuttal in camera-ready, especially regarding justifications on orthonormality assumptions and discussions on limitations.

**Justification For Why Not Higher Score:**

The theoretical analysis is based on some strong assumptions such as the orthonormality of embedding vectors and stationary gradients, which does not exactly hold in practice yet is aligned with empirical observations to some degree.

**Justification For Why Not Lower Score:**

N/A

---

### Decision · Program_Chairs · 2024-01-16

Accept (poster)